# Robust Aggregation Operators for Intuitionistic Fuzzy Hypersoft Set with Their Application to Solve MCDM Problem

**DOI:** 10.3390/e23060688

**Published:** 2021-05-29

**Authors:** Rana Muhammad Zulqarnain, Imran Siddique, Rifaqat Ali, Dragan Pamucar, Dragan Marinkovic, Darko Bozanic

**Affiliations:** 1Department of Mathematics, University of Management and Technology, Sialkot Campus, Lahore 51310, Pakistan; 2Department of Mathematics, University of Management and Technology, Lahore 54770, Pakistan; 3Department of Mathematics, College of Science and Arts, Muhayil, King Khalid University, Abha 61413, Saudi Arabia; rrafat@kku.edu.sa; 4Department of Logistics, Military academy, University of Defence in Belgarde, Pavla Jurisica Sturma 33, 11000 Belgrade, Serbia; dragan.pamucar@va.mod.gov.rs; 5Faculty of Mechanical and Transport Systems, Technische Universität Berlin, 10623 Berlin, Germany; dragan.marinkovic@tu-berlin.de; 6Military Academy, University of Defence in Belgarde, Pavla Jurisica Sturma 33, 11000 Belgrade, Serbia; dbozanic@yahoo.com

**Keywords:** hypersoft set, intuitionistic fuzzy soft set, intuitionistic fuzzy hypersoft set, IFHSWA operator, IFHSWG operator, SSCM

## Abstract

In this paper, we investigate the multi-criteria decision-making complications under intuitionistic fuzzy hypersoft set (IFHSS) information. The IFHSS is a proper extension of the intuitionistic fuzzy soft set (IFSS) which discusses the parametrization of multi-sub attributes of considered parameters, and accommodates more hesitation comparative to IFSS utilizing the multi sub-attributes of the considered parameters. The main objective of this research is to introduce operational laws for intuitionistic fuzzy hypersoft numbers (IFHSNs). Additionally, based on developed operational laws two aggregation operators (AOs), i.e., intuitionistic fuzzy hypersoft weighted average (IFHSWA) and intuitionistic fuzzy hypersoft weighted geometric (IFHSWG), operators have been presented with their fundamental properties. Furthermore, a decision-making approach has been established utilizing our developed aggregation operators (AOs). Through the established approach, a technique for solving decision-making (DM) complications is proposed to select sustainable suppliers in sustainable supply chain management (SSCM). Moreover, a numerical description is presented to ensure the validity and usability of the proposed technique in the DM process. The practicality, effectivity, and flexibility of the current approach are demonstrated through comparative analysis with the assistance of some prevailing studies.

## 1. Introduction

Decision-making is an interesting concern to select the perfect alternative for any particular purpose. Firstly, it is supposed that details about alternatives are accumulated in crisp numbers, but in real-life situations, collective farm information always contains wrong and inaccurate information. Fuzzy sets [1] are similar to sets having an element of membership (Mem) degree. In classical set theory, the Mem degree of the elements in a set is examined in binary form to see that the element is not entirely concomitant to the set. In contrast, the fuzzy set theory enables advanced Mem categorization of the components in the set. It is portrayed by the Mem function, and also the multipurpose unit interval of the Mem function is [0, 1]. In some circumstances, decision-makers consider the Mem and non-membership (Nmem) values of objects. In such cases Zadeh’s FS is unable to handle the imprecise and vague information. Atanassov [2] established the notion of intuitionistic fuzzy sets (IFS) to deal with the above-mentioned concerns. In addition, several other theories have been proposed to overcome such complications, such as cubic IFS [3], interval-valued IFS [4], entropy and distance measures for IFSS [5], etc.

Atanassov’s IFS accommodates the inadequate information utilizing Mem and NMem values, but IFS is unable to deal with incompatible and inexact data in which alternatives have parametric values. To overcome such complications, Molodtsov [6] presented the soft set (SS) for indeterminate, uncertain, and vague substances. Maji et al. [7] established the notion of the fuzzy soft set (FSS) by merging FS and SS. Ali et al. [8] proposed a novel decision-making approach for bipolar FSS utilizing different types of parameter reduction. Maji et al. [9] developed the notion of IFSS and presented some fundamental operations with their desirable properties. Garg and Arora [10] introduced the correlation coefficient for IFSS, and extended the TOPSIS method for IFSS. They also utilized their developed TOPSIS technique to solve multi-attribute decision-making (MADM) obstacles. Wei and Zhang [11] presented two novel entropy measures for IFS and interval-valued IFS based on cosine function with their fundamental properties and established a decision-making technique to utilize their developed entropy measures. Zulqarnain et al. [12] proposed the correlation coefficient for IVFSS and developed the TOPSIS technique, utilizing their presented correlation coefficient. They also introduced the AOs for IVFSS, and used their established techniques to solve decision-making complications. Garg and Arora [13] extended the notion of IFSS, proposed the most generalized form of IFSS, and developed some AOs with their fundamental properties.

Wang et al. [14] extended the PFSs and introduced interactive Hamacher operations with some novel AOs. They also established a DM method to solve MADM problems by using their proposed operators. Wang and Li [15] extended the notion of PFSs to interval-valued PFSs with some desirable operators. They also developed a DM technique utilizing their proposed operators to solve multi-attribute group decision-making (MAGDM) problems. Wang et al. [16] established a MADM approach utilizing interval-valued q-rung orthopair 2-tuple linguistic. Pamucar [17] established an MCDM approach by merging two existing studies of interval grey numbers and normalized weighted geometric Dombi–Bonferroni mean operators. Peng and Yuan [18] established some novel operators, such as Pythagorean fuzzy point operators, and developed a DM technique using their proposed operators. Garg [19] extended the weighted AOs to PFSs and acquired numerous operators, introducing a DM technique founded on settled operators. Wang and Liu [20] studied a intuitionistic fuzzy Einstein weighted geometric operator and established a DM approach to solving DM complications. Garg [21] developed the logarithmic operational laws for PFSs and proposed some AOs. Wang and Liu [22] introduced the intuitionistic fuzzy Einstein weighted averaging operator and the intuitionistic fuzzy Einstein ordered weighted averaging operator with their desirable properties. Arora and Garg [23] presented the operational laws for linguistic IFS and developed prioritized AOs. 

Faizi et al. [24] introduced two novel techniques to obtain the best priority vector for the solution of MAGDM problems utilizing intuitionist 2-tuple linguistic sets. Based on Hamacher operations, they presented some operational laws in an under-considered environment. Xu [25] presented some innovative AOs for IFSs and proposed the comparison laws for IFNs. Sinani et al. [26] extended the Hamy mean and Dombi operators for rough numbers and established a DM approach to resolving MADM challenges based on their acquired operators. Riaz et al. [27] introduced the soft multi-set topology on a soft multi-set with aggregation operators and the MCDM technique. Peng et al. [28] presented the notion of PFSS with its fundamental properties merging two prevailing techniques, PFS and SS. Athira et al. [29] extended the concept of PFSS, and introduced entropy measure under-considered environment. Zulqarnain et al. [30] established operational laws for Pythagorean fuzzy soft numbers (PFSNs) and developed AOs, such as Pythagorean fuzzy soft weighted average and geometric, by using defined operational laws for PFSNs. They also planned a DM approach to solve MADM problems with the help of presented operators. Riaz et al. [31] have established AOs utilizing Einstein Operations and examined their fundamental properties; they also proposed a DM technique to solve MCDM obstacles. Faizi et al. [32] extended the notion of normalized interval-valued triangular fuzzy numbers, and developed an MCDM technique to solve DM issues. Several techniques have been established to solve multi-criteria decision analysis (MCDA) which provide suitable results in real-life complications [33,34]. Salabun et al. [35] worked to benchmark selected MCDA techniques and determined that a set of realistic MCDA approaches. Referring to the mentioned guide, they propose to organize simulation experimentations.

The existing studies only deal with inadequate information due to membership and non-membership values. However, these theories cannot handle the overall incompatible, as well as imprecise, information. When any attribute from a set of parameters comprises further sub-attributes, the prevailing theories fail to solve such types of problems. To overcome the aforementioned limitations, Smarandache [36] developed the idea of SS to hypersoft sets (HSS) by substituting the one-parameter function f to a multi-parameter (sub-attribute) function. Samarandache claimed that the established HSS can competently deal with uncertain objects, in comparison to SS. Nowadays, HSS theory and its extensions have been gaining unexpected traction. Several investigators have examined progressed distinctive operators along with characteristics under HSS and its extensions [37,38,39,40,41,42,43]. Zulqarnain et al. [44] presented the IFHSS, which is the generalized version of IFSS. They established the TOPSIS method to resolve the MADM problem, utilizing the developed correlation coefficient.

The essential objective of the following scientific research is to grow novel AOs for the IFHSS environment and processing mechanism, which can also follow the assumptions of IFHSNs. Furthermore, we developed an algorithm to explain the MCDM problem, and presented a numerical illustration to justify the effectiveness of the proposed approach under the IFHSS environment. Supplier selection and evaluation is a critical factor in business activity. Recent government policy changes have led to supplier selection being considered from various perspectives, including environmental and social imperatives. Thus, in the literature, the problem is referred to as sustainable supplier selection and described as a problem for MCDM. Simultaneously, many papers [45,46,47,48] point to the need for further research using MCDM methods in sustainable supplier selection, oriented at properly reflecting uncertainties in the data of the environment and the preferences of the decision-maker. To improve the computing power and flexibility of IFSS, we first summarize the decision formula to incorporate the views of decision-makers into IFHSS terms, and propose operational rules for IFHSS. According to the newly developed operational rules, there are two AOs, namely IFHSWA and IFHSWG operators, which have been established. Many of the associated properties of these operators are also inspected. The score function and accuracy function of IHFSS are also discussed to compare IFHSS. The algorithmic rule following the proposed operator to resolve the DM problems is anticipated, along with a numerical example used to demonstrate the effectiveness of the introduced DM approach.

The rest of the study is planned thus: In Section 2, we present some initial impressions, such as SS, HSS, and IFHSS, which help us to build a successful investigation structure. We formulated some operational laws for IFHSNs in accordance with established operating laws, and presented the aggregation operators (AOs) with their properties in Section 3. In Section 4, a DM method is developed for SSCM by using the proposed operator. To ensure the practicality of the established DM approach a numerical example is presented. In addition, we utilize some available techniques to indicate analysis within our own designed technique. Furthermore, we give the benefits of planned algorithms, simplicity, flexibility, and effectivity. In Section 5, we will concisely discuss and compare existing techniques along with the projected approach.

## 2. Preliminaries

In the following section, we are going to review some fundamental definitions that support us in establishing the following research structures, such as SS, HSS, FHSS, as well as IFHSS.

**Definition** **1.**
*[6] Let*
U
*be the universal set and*
ℰ
*be the set of attributes concerning*
U
*. Let*
P(U)
*be the power set of*
U
*and*
C⊆ℰ
*. A pair (*
ℱ,C
*) is called a SS over*
U
*and its mapping is given as*
ℱ: C→P(U)

*It is also defined as:*
(ℱ,C)={ℱ(e)∈P(U):e∈ℰ,ℱ(e)=∅ if e∉C}


**Definition** **2.***[7] ℱ(U) be a collection of all fuzzy subsets over*U*and*ℰ*be a set of attributes. Let*C ⊆ ℰ*, then a pair (*ℱ,C*) is called FSS over*U*, where*ℱ*is a mapping such as*ℱ: C→ 𝘍(U)

**Definition** **3.***[36] Let*U*be a universe of discourse and*P(U)*be a power set of*U*and*k*= {*$k_{1}$*,*$k_{2}$*,*$k_{3}$*,…,*$\;k_{n}$*},(n ≥ 1) be a set of attributes and set*$K_{i}$*a set of corresponding sub-attributes of*$k_{i}$*respectively with*$K_{i}$*∩*$K_{j}$*= φ for**n**≥ 1 for each**i**,**j ε**{1,2,3 …**n**} and**i**≠**j**. Assume*$K_{1}$*×*$K_{2}$*×*$K_{3}$*× … ×*$K_{n}$*=*$\dddot{C}$*=*$\left\{c_{1h}\times c_{2k}\times \ldots \times c_{nl}\right\}$*be a collection of multi-attributes, where 1*≤ h ≤ α*, 1*≤ k ≤ β*, and 1*≤ l ≤ γ*, and*α*,*β*, and*γ ∈*ℕ. Then the pair (*ℱ*,*$K_{1}$*×*$K_{2}$*×*$K_{3}$*× … ×*$K_{n}$*=*$\dddot{C}$*) is said to be HSS over*U*and its mapping is defined as*$$ℱ:\;K_{1}\times \;K_{2}\times \;K_{3}\times \ldots \times K_{n}=\;\dddot{C}\to P\left(U\right)$$
*It is also defined as*
$$(ℱ,\dddot{C})=\left\{\left(\overset{ˇ}{a},\;{ℱ}_{\dddot{A}}\left(\overset{ˇ}{a}\right)\right):\;\overset{ˇ}{a}\in \dddot{A},\;{ℱ}_{\dddot{A}}\left(\overset{ˇ}{a}\right)\in P(U)\right\}$$


**Definition** **4.***[44] Let*U* be a universe of discourse and*P(U)*be a power set of*U*and*k*= {*$k_{1}$*,*$k_{2}$*,*$k_{3}$*,…,*$\;k_{n}$*},(n ≥ 1) be a set of attributes and set*$K_{i}$*a set of corresponding sub-attributes of*$k_{i}$*respectively with*$K_{i}$*∩*$K_{j}$*= φ for**n**≥ 1 for each**i**,**j ε**{1,2,3 …**n**} and**i**≠**j**. Assume*$K_{1}$*×*$K_{2}$*×*$K_{3}$*× … ×*$K_{n}$*=*$\dddot{C}$*=*$\left\{c_{1h}\times c_{2k}\times \cdots \times c_{nl}\right\}$*be a collection of sub-attributes, where 1*≤ h ≤ α*, 1*≤ k ≤ β*, and 1*≤ l ≤ γ*, and*α*,*β*, and*γ ∈*ℕ and*$IFS^{U}$*be a collection of all intuitionistic fuzzy subsets over*U*. Then the pair (*ℱ*,*$K_{1}$*×*$K_{2}$*×*$K_{3}$*× … ×*$K_{n}$*=*$\dddot{C}$*) is said to be IFHSS over*U,*and its mapping is defined as*$$ℱ:\;K_{1}\times \;K_{2}\times \;K_{3}\times \ldots \times K_{n}\;\;\to \;IFS^{U}.$$*It is also defined as*$$(ℱ,\dddot{C})=\left\{\left(\overset{ˇ}{c},\;{ℱ}_{\dddot{C}}\left(\overset{ˇ}{c}\right)\right):\;\overset{ˇ}{c}\in \dddot{C},\;{ℱ}_{\dddot{C}}\left(\overset{ˇ}{c}\right)\in IFS^{U}\in \;\left[0,\;1\right]\right\}$$*where*${ℱ}_{\dddot{C}}\left(\overset{ˇ}{c}\right)$*=*$\left\{\delta ,\;\mu_{ℱ\left(\overset{ˇ}{c}\right)}\left(\delta \right),\;\vartheta_{ℱ\left(\overset{ˇ}{c}\right)}\left(\delta \right):\;\delta \in U\right\}$*, where*$\mu_{ℱ\left(\overset{ˇ}{c}\right)}\left(\delta \right)$*and*$\vartheta_{ℱ\left(\overset{ˇ}{c}\right)}\left(\delta \right)$*represents the MD and NMD of the sub-attributes of the considered parameters such as*$\mu_{ℱ\left(\overset{ˇ}{c}\right)}\left(\delta \right)$*,*$\vartheta_{ℱ\left(\overset{ˇ}{c}\right)}\left(\delta \right)$ ∈ [0, 1]*, and 0*≤ $\mu_{ℱ\left(\overset{ˇ}{c}\right)}\left(\delta \right)$*+*$\vartheta_{ℱ\left(\overset{ˇ}{c}\right)}\left(\delta \right)\leq$*1.*

**Remark** **1.***If*$\mu_{ℱ\left(\overset{ˇ}{c}\right)}\left(\delta \right)+\vartheta_{ℱ\left(\overset{ˇ}{c}\right)}\left(\delta \right)$ ≤*1 is held and all parameters of a set of attributes have no further sub-attribute. Then, IFHSS was reduced to IFSS [9].*

The IFHSN ${ℱ}_{\delta_{i}}\left({\overset{ˇ}{c}}_{j}\right)$ = $\left\{\left(\mu_{ℱ\left({\overset{ˇ}{c}}_{j}\right)}\left(\delta_{i}\right),\;\vartheta_{ℱ\left({\overset{ˇ}{c}}_{j}\right)}\left(\delta_{i}\right)\right)\;|\delta_{i}\in U\right\}$ can be express as $J_{{\overset{ˇ}{c}}_{ij}}$ = $\left\langle \mu_{ℱ\left({\overset{ˇ}{c}}_{ij}\right)},\;\vartheta_{ℱ\left({\overset{ˇ}{c}}_{ij}\right)}\right\rangle$ for readers’ convenience. To rank the alternatives scoring function of $J_{{\overset{ˇ}{c}}_{ij}}$ is defined in the following:(1)$$S\left(J_{{\overset{ˇ}{c}}_{ij}}\right)=\mu_{ℱ\left({\overset{ˇ}{c}}_{ij}\right)}-\vartheta_{ℱ\left(c_{ij}\right)},\;S\left(J_{{\overset{ˇ}{c}}_{ij}}\right)\in \left[-1,\;1\right]$$

However, sometimes the scoring function such as $J_{{\overset{ˇ}{c}}_{11}}$ = ⟨0.4, 0.7⟩ and $J_{{\overset{ˇ}{c}}_{12}}$ = ⟨0.5,0.8⟩ is unable to compute the two IFHSNs. In such cases it can be difficult to decide which value is most suitable $S\left(J_{{\overset{ˇ}{c}}_{11}}\right)$ = 0.3 =$\;S\left(J_{{\overset{ˇ}{c}}_{12}}\right)$. Accuracy function has been introduced to overcome such difficulties:(2)$$H\left(J_{{\overset{ˇ}{c}}_{ij}}\right)=\mu_{ℱ\left({\overset{ˇ}{c}}_{ij}\right)}+\vartheta_{ℱ\left({\overset{ˇ}{c}}_{ij}\right)},\;H\left(J_{{\overset{ˇ}{c}}_{ij}}\right)\in \left[0,\;1\right].$$

Thus, to compare two IFHSNs $J_{{\overset{ˇ}{c}}_{ij}}$ and $T_{{\overset{ˇ}{c}}_{ij}}$, the subsequent ranking and comparison laws are classified as follows:
If $S\left(J_{{\overset{ˇ}{c}}_{ij}}\right)$ > $S\left(T_{c_{ij}}\right)$, then $J_{{\overset{ˇ}{c}}_{ij}}$ > $T_{{\overset{ˇ}{c}}_{ij}}$.If $S\left(J_{{\overset{ˇ}{c}}_{ij}}\right)$ = $S\left(T_{{\overset{ˇ}{c}}_{ij}}\right)$, then
oIf $H\left(J_{{\overset{ˇ}{c}}_{ij}}\right)$ > $H\left(T_{{\overset{ˇ}{c}}_{ij}}\right)$, then $J_{{\overset{ˇ}{c}}_{ij}}$ > $T_{{\overset{ˇ}{c}}_{ij}}$oIf $H\left(J_{{\overset{ˇ}{c}}_{ij}}\right)$ = $H\left(T_{{\overset{ˇ}{c}}_{ij}}\right)$, then $J_{{\overset{ˇ}{c}}_{ij}}$ = $T_{{\overset{ˇ}{c}}_{ij}}$.



## 3. Aggregation Operators for Intuitionistic Fuzzy Hypersoft Numbers

In this section, we present the operational laws for IFHSNs and propose the IFHSWA, and IFHSWG operators for IFHSS. Furthermore, we discuss the fundamental properties of IFHSWA and IFHSWG operators utilizing our developed IFHSNs.

### 3.1. Operational Laws for Intuitionistic Fuzzy Hypersoft Numbers

**Definition** **5.**
*Let*
$J_{{\overset{ˇ}{c}}_{k}}$
*=*
$\left(\mu_{{\overset{ˇ}{c}}_{k}},\;\vartheta_{{\overset{ˇ}{c}}_{k}}\right)$
*,*
$J_{{\overset{ˇ}{c}}_{11}}$
*=*
$\left(\mu_{{\overset{ˇ}{c}}_{11}},\;\vartheta_{{\overset{ˇ}{c}}_{11}}\right)$
*, and*
$J_{{\overset{ˇ}{c}}_{12}}$
*=*
$\left(\mu_{{\overset{ˇ}{c}}_{12}},\;\vartheta_{{\overset{ˇ}{c}}_{12}}\right)$
*be three IFHSNs and*
α
*be a positive real number, by algebraic norms, we have*
$J_{{\overset{ˇ}{c}}_{11}}⊕$ $J_{{\overset{ˇ}{c}}_{12}}$ = $\left\langle \mu_{{\overset{ˇ}{c}}_{11}}+\mu_{{\overset{ˇ}{c}}_{12}}-\mu_{{\overset{ˇ}{c}}_{11}}\mu_{{\overset{ˇ}{c}}_{12}},\;\vartheta_{{\overset{ˇ}{c}}_{11}}\vartheta_{{\overset{ˇ}{c}}_{12}}\right\rangle$$J_{{\overset{ˇ}{c}}_{11}}⊗J_{c_{12}}$ = $\left\langle \mu_{{\overset{ˇ}{c}}_{11}}\mu_{{\overset{ˇ}{c}}_{12}},\;\vartheta_{{\overset{ˇ}{c}}_{11}}+\vartheta_{{\overset{ˇ}{c}}_{12}}-\vartheta_{{\overset{ˇ}{c}}_{11}}\vartheta_{{\overset{ˇ}{c}}_{12}}\right\rangle$$\alpha J_{{\overset{ˇ}{c}}_{k}}$ = $\left\langle \left[1-{\left(1-\mu_{{\overset{ˇ}{c}}_{k}}\right)}^{\alpha },\;{\left(\;\vartheta_{{\overset{ˇ}{c}}_{k}}\right)}^{\alpha }\right]\right\rangle$$J_{{\overset{ˇ}{\overset{ˇ}{c}}}_{k}}^{\alpha }$ = $\left\langle \left[{\left(\mu_{{\overset{ˇ}{c}}_{k}}\right)}^{\alpha },\;1-{\left(1-\vartheta_{{\overset{ˇ}{c}}_{k}}\right)}^{\alpha }\right]\right\rangle$
*Some average and geometric AOs for IFHSSs are described based on the above rules for collecting IFHSNs*Δ. 

**Definition** **6.***Let*$J_{{\overset{ˇ}{c}}_{k}}$*=*$\left(\mu_{{\overset{ˇ}{c}}_{k}},\;\vartheta_{{\overset{ˇ}{c}}_{k}}\right)$*be an IFHSN,*${Ω}_{i}$*and*$\gamma_{j}$*are weight vector for expert’s and sub-attributes of selected parameters respectively with given conditions*${Ω}_{i}$ >*0,*$\sum_{i=1}^{n}{Ω}_{i}$*= 1,*$\gamma_{j}$ >*0,*$\sum_{\;j=1}^{m}\gamma_{j}$*= 1. Then IFHSWA operator is defined as**IFHSWA:*$\Delta^{n}$ → Δ*defined as follows:*(3)$$IFHSWA\;\left(J_{{\overset{ˇ}{c}}_{11}},\;J_{{\overset{ˇ}{c}}_{12}},\ldots ,\;J_{{\overset{ˇ}{c}}_{nm}}\right)={⊕}_{\;j=1}^{m}\gamma_{j}\left({⊕}_{i=1}^{n}{Ω}_{i}J_{{\overset{ˇ}{c}}_{ij}}\;\right).$$

**Theorem** **1.***Let*$J_{{\overset{ˇ}{c}}_{k}}$*=*$\left(\mu_{{\overset{ˇ}{c}}_{k}},\;\vartheta_{{\overset{ˇ}{c}}_{k}}\right)$*be an IFHSN. Then, the aggregated values obtained by using Equation (3) is also an IFHSN and*(4)$$IFHSWA\;\left(J_{{\overset{ˇ}{c}}_{11}},\;J_{{\overset{ˇ}{c}}_{12}},\ldots ,\;J_{{\overset{ˇ}{c}}_{nm}}\right)\;=\langle 1-\prod_{j=1}^{m}{\left(\prod_{i=1}^{n}{\left(1-\mu_{{\overset{ˇ}{c}}_{ij}}\right)}^{{Ω}_{i}}\right)}^{\gamma_{j}},\;\prod_{j=1}^{m}{\left(\prod_{i=1}^{n}{\left(\;\vartheta_{{\overset{ˇ}{c}}_{ij}}\right)}^{{Ω}_{i}}\right)}^{\gamma_{j}}\rangle .$$*where*${Ω}_{i}$*and*$\gamma_{j}$*are weight vector for expert’s and sub-attributes of the parameters correspondingly with given circumstances*${Ω}_{i}$ >*0,*$\sum_{i=1}^{n}{Ω}_{i}$*= 1,*$\gamma_{j}$ >*0,*$\sum_{\;j=1}^{m}\gamma_{j}$*= 1.*

**Proof.** We can prove this by applying the principle of mathematical induction such as follows:For n=1, we get ${Ω}_{1}$ = 1. Then, we have
$$\begin{array}{l} IFHSWA\;\left(J_{{\overset{ˇ}{c}}_{11}},\;J_{{\overset{ˇ}{c}}_{12}},\ldots ,\;J_{{\overset{ˇ}{c}}_{nm}}\right)={⊕}_{\;j=1}^{m}\gamma_{j}J_{{\overset{ˇ}{c}}_{ij}}\\ IFHSWA\;\left(J_{{\overset{ˇ}{c}}_{11}},\;J_{{\overset{ˇ}{c}}_{12}},\ldots ,\;J_{{\overset{ˇ}{c}}_{nm}}\right)=\langle 1-\prod_{j=1}^{m}{\left(1-\mu_{{\overset{ˇ}{c}}_{1j}}{}^{2}\right)}^{\gamma_{j}},\;\prod_{j=1}^{m}{\left(\vartheta_{{\overset{ˇ}{c}}_{1j}}\right)}^{\gamma_{j}}\rangle \\ =\langle 1-\prod_{j=1}^{m}{\left(\prod_{i=1}^{1}{\left(1-\mu_{{\overset{ˇ}{d}}_{1j}}{}^{2}\right)}^{{Ω}_{i}}\right)}^{\gamma_{j}},\;\prod_{j=1}^{m}{\left(\prod_{i=1}^{1}{\left(\vartheta_{{\overset{ˇ}{d}}_{1j}}\right)}^{{Ω}_{i}}\right)}^{\gamma_{j}}\rangle . \end{array}$$For m=1, we get $\gamma_{1}$ = 1. Then, we have
$$\begin{array}{l} IFHSWA\;\left(J_{{\overset{ˇ}{c}}_{11}},\;J_{{\overset{ˇ}{c}}_{12}},\ldots ,\;J_{{\overset{ˇ}{c}}_{nm}}\right)={⊕}_{\;i=1}^{n}{Ω}_{i}J_{{\overset{ˇ}{c}}_{ij}}\\ =\langle 1-\prod_{i=1}^{n}{\left(1-\mu_{{\overset{ˇ}{c}}_{i1}}{}^{2}\right)}^{{Ω}_{i}},\;\prod_{i=1}^{n}{\left(\vartheta_{{\overset{ˇ}{c}}_{i1}}\right)}^{{Ω}_{i}}\rangle \\ =\langle 1-\prod_{j=1}^{1}{\left(\prod_{i=1}^{n}{\left(1-\mu_{{\overset{ˇ}{c}}_{ij}}{}^{2}\right)}^{{Ω}_{i}}\right)}^{\gamma_{j}},\;\prod_{j=1}^{1}{\left(\prod_{i=1}^{n}{\left(\vartheta_{{\overset{ˇ}{c}}_{ij}}\right)}^{{Ω}_{i}}\right)}^{\gamma_{j}}\rangle . \end{array}$$This shows that Equation (4) satisfies for n=1 and m=1. Consider Equation (4) holds for $m=\beta_{1}+1$, $n=\beta_{2}$ and $m=\beta_{1}$, $n=\beta_{2}+1$, such as:$$\begin{array}{l} {⊕}_{\;j=1}^{\beta_{1}+1}\gamma_{j}\left({⊕}_{i=1}^{\beta_{2}}{Ω}_{i}J_{{\overset{ˇ}{c}}_{ij}}\;\right)=\langle 1-\prod_{j=1}^{\beta_{1}+1}{\left(\prod_{i=1}^{\beta_{2}}{\left(1-\mu_{{\overset{ˇ}{c}}_{ij}}{}^{2}\right)}^{{Ω}_{i}}\right)}^{\gamma_{j}},\;\prod_{j=1}^{\beta_{1}+1}{\left(\prod_{i=1}^{\beta_{2}}{\left(\vartheta_{{\overset{ˇ}{c}}_{ij}}\right)}^{{Ω}_{i}}\right)}^{\gamma_{j}}\rangle \\ {⊕}_{\;j=1}^{\beta_{1}}\gamma_{j}\left({⊕}_{i=1}^{\beta_{2}+1}{Ω}_{i}J_{{\overset{ˇ}{c}}_{ij}}\right)=\langle 1-\prod_{j=1}^{\beta_{1}}{\left(\prod_{i=1}^{\beta_{2}+1}{\left(1-\mu_{{\overset{ˇ}{c}}_{ij}}{}^{2}\right)}^{{Ω}_{i}}\right)}^{\gamma_{j}},\;\prod_{j=1}^{\beta_{1}}{\left(\prod_{i=1}^{\beta_{2}+1}{\left(\vartheta_{{\overset{ˇ}{c}}_{ij}}\right)}^{{Ω}_{i}}\right)}^{\gamma_{j}}\rangle \end{array}$$For $m=\beta_{1}+1$ and $n=\beta_{2}+1$, we have
$$\begin{array}{l} {⊕}_{\;j=1}^{\beta_{1}+1}\gamma_{j}\left({⊕}_{i=1}^{\beta_{2}+1}{Ω}_{i}J_{{\overset{ˇ}{c}}_{ij}}\;\right)={⊕}_{\;j=1}^{\beta_{1}+1}\gamma_{j}\left({⊕}_{i=1}^{\beta_{2}}{Ω}_{i}J_{{\overset{ˇ}{c}}_{ij}}⊕{Ω}_{\beta_{2}+1}J_{{\overset{ˇ}{c}}_{\left(\beta_{2}+1\right)j}}\;\right)\\ ={⊕}_{\;j=1}^{\beta_{1}+1}{⊕}_{i=1}^{\beta_{2}}\gamma_{j}{Ω}_{i}J_{{\overset{ˇ}{c}}_{ij}}{⊕}_{\;j=1}^{\beta_{1}+1}\gamma_{j}{Ω}_{\beta_{2}+1}J_{{\overset{ˇ}{c}}_{\left(\beta_{2}+1\right)j}}\\ =\langle \begin{array}{c} 1-\prod_{j=1}^{\beta_{1}+1}{\left(\prod_{i=1}^{\beta_{2}}{\left(1-\mu_{{\overset{ˇ}{c}}_{ij}}{}^{2}\right)}^{{Ω}_{i}}\right)}^{\gamma_{j}}⊕1-\prod_{j=1}^{\beta_{1}+1}{\left({\left(1-\mu_{{\overset{ˇ}{c}}_{\left(\beta_{2}+1\right)j}}{}^{2}\right)}^{{Ω}_{\beta_{2}+1}}\right)}^{\gamma_{j}},\\ \prod_{j=1}^{\beta_{1}+1}{\left(\prod_{i=1}^{\beta_{2}}{\left(\vartheta_{{\overset{ˇ}{c}}_{ij}}\right)}^{{Ω}_{i}}\right)}^{\gamma_{j}}⊕\prod_{j=1}^{\beta_{1}+1}{\left({\left(\vartheta_{{\overset{ˇ}{c}}_{\left(\beta_{2}+1\right)j}}\right)}^{{Ω}_{\beta_{2}+1}}\right)}^{\gamma_{j}} \end{array}\rangle \\ =\begin{array}{c} \left\langle 1-\prod_{j=1}^{\beta_{1}+1}{\left(\prod_{i=1}^{\beta_{2}+1}{\left(1-\mu_{{\overset{ˇ}{c}}_{ij}}{}^{2}\right)}^{{Ω}_{i}}\right)}^{\gamma_{j}},\;\prod_{j=1}^{\beta_{1}+1}{\left(\prod_{i=1}^{\beta_{2}+1}{\left(\vartheta_{{\overset{ˇ}{c}}_{ij}}\right)}^{{Ω}_{i}}\right)}^{\gamma_{j}}\right\rangle \end{array}. \end{array}$$Hence, it is true for $m=\beta_{1}+1$ and $n=\beta_{2}+1$. □

### 3.2. Properties of IFHSWA Operator 

#### 3.2.1. (Idempotency) 

If $J_{{\overset{ˇ}{c}}_{ij}}$ = $J_{{\overset{ˇ}{c}}_{ij}}$ = $\left(\mu_{{\overset{ˇ}{c}}_{ij}},\;\vartheta_{{\overset{ˇ}{c}}_{ij}}\right)$ ∀ i, j, then, 

$IFHSWA\;\left(J_{{\overset{ˇ}{c}}_{11}},\;J_{{\overset{ˇ}{c}}_{12}},\ldots ,\;J_{{\overset{ˇ}{c}}_{nm}}\right)$ = $J_{\overset{ˇ}{c}}$.

**Proof.** As we know $J_{{\overset{ˇ}{c}}_{ij}}$ = $J_{\overset{ˇ}{c}}$ = $\left(\mu_{{\overset{ˇ}{c}}_{ij}},\;\vartheta_{{\overset{ˇ}{c}}_{ij}}\right)$ to be a collection of IFHSNs, then by using Equation (4).
$$\begin{array}{l} IFHSWA\;\;\left(J_{{\overset{ˇ}{c}}_{11}},\;J_{{\overset{ˇ}{c}}_{12}},\ldots ,\;J_{{\overset{ˇ}{c}}_{nm}}\right)=\langle 1-\prod_{j=1}^{m}{\left(\prod_{i=1}^{n}{\left(1-\mu_{{\overset{ˇ}{c}}_{ij}}\right)}^{{Ω}_{i}}\right)}^{\gamma_{j}},\;\prod_{j=1}^{m}{\left(\prod_{i=1}^{n}{\left(\vartheta_{{\overset{ˇ}{c}}_{ij}}\right)}^{{Ω}_{i}}\right)}^{\gamma_{j}}\rangle \\ =\langle 1-{\left({\left(1-\mu_{{\overset{ˇ}{c}}_{ij}}\right)}^{\sum_{i=1}^{n}{Ω}_{i}}\right)}^{\sum_{j=1}^{m}\gamma_{j}},\;{\left({\left(\vartheta_{{\overset{ˇ}{c}}_{ij}}\right)}^{\sum_{i=1}^{n}{Ω}_{i}}\right)}^{\sum_{j=1}^{m}\gamma_{j}}\rangle \\ =\langle 1-\left(1-\mu_{{\overset{ˇ}{c}}_{ij}}\right),\;\vartheta_{{\overset{ˇ}{c}}_{ij}}\rangle =\left(\mu_{{\overset{ˇ}{c}}_{ij}},\;\vartheta_{{\overset{ˇ}{c}}_{ij}}\right)=J_{\overset{ˇ}{c}}. \end{array}$$Which completes the proof. □

#### 3.2.2. (Boundedness) 

Let $J_{{\overset{ˇ}{c}}_{ij}}$ be a collection of IFHSNs and

Jcˇij− = ⟨minjmini{μcˇij}, maxjmaxi{ϑcˇij}⟩ and Jcˇij+ = ⟨maxjmaxi{μcˇij}, minjmini{ϑcˇij}⟩, then

$J_{{\overset{ˇ}{c}}_{ij}}{}^{-}$ ≤ $IFHSWA\;\left(J_{{\overset{ˇ}{c}}_{11}},\;J_{{\overset{ˇ}{c}}_{12}},\ldots ,\;J_{{\overset{ˇ}{c}}_{nm}}\right)$ ≤ $J_{{\overset{ˇ}{c}}_{ij}}{}^{+}$.

**Proof.** As we know $J_{{\overset{ˇ}{c}}_{ij}}$ = $\left(\mu_{{\overset{ˇ}{c}}_{ij}},\;J_{{\overset{ˇ}{c}}_{ij}}\right)$ to be an IFHSN, then
(5)$$\begin{array}{l} \begin{array}{c} min\\ j \end{array}\begin{array}{c} min\\ i \end{array}\left\{\mu_{{\overset{ˇ}{c}}_{ij}}\right\}\leq \mu_{{\overset{ˇ}{c}}_{ij}}\leq \begin{array}{c} max\\ j \end{array}\begin{array}{c} max\\ i \end{array}\left\{\mu_{{\overset{ˇ}{c}}_{ij}}\right\}\\ \Rightarrow 1-\begin{array}{c} max\\ j \end{array}\begin{array}{c} max\\ i \end{array}\left\{\mu_{{\overset{ˇ}{c}}_{ij}}\right\}\leq 1-\mu_{{\overset{ˇ}{c}}_{ij}}\leq 1-\begin{array}{c} min\\ j \end{array}\begin{array}{c} min\\ i \end{array}\left\{\mu_{{\overset{ˇ}{c}}_{ij}}\right\}\\ \Leftrightarrow {\left(1-\begin{array}{c} max\\ j \end{array}\begin{array}{c} max\\ i \end{array}\left\{\mu_{{\overset{ˇ}{c}}_{ij}}\right\}\right)}^{{Ω}_{i}}\leq {\left(1-\mu_{{\overset{ˇ}{c}}_{ij}}\right)}^{{Ω}_{i}}\leq {\left(1-\begin{array}{c} min\\ j \end{array}\begin{array}{c} min\\ i \end{array}\left\{\mu_{{\overset{ˇ}{c}}_{ij}}\right\}\right)}^{{Ω}_{i}}\\ \Leftrightarrow {\left(1-\begin{array}{c} max\\ j \end{array}\begin{array}{c} max\\ i \end{array}\left\{\mu_{{\overset{ˇ}{c}}_{ij}}\right\}\right)}^{\sum_{i=1}^{n}{Ω}_{i}}\leq \prod_{i=1}^{n}{\left(1-\mu_{{\overset{ˇ}{c}}_{ij}}\right)}^{{Ω}_{i}}\leq {\left(1-\begin{array}{c} min\\ j \end{array}\begin{array}{c} min\\ i \end{array}\left\{\mu_{{\overset{ˇ}{c}}_{ij}}\right\}\right)}^{\sum_{i=1}^{n}{Ω}_{i}}\\ \Leftrightarrow {\left(1-\begin{array}{c} max\\ j \end{array}\begin{array}{c} max\\ i \end{array}\left\{\mu_{{\overset{ˇ}{c}}_{ij}}\right\}\right)}^{\sum_{j=1}^{m}\gamma_{j}}\leq \prod_{j=1}^{m}{\left(\prod_{i=1}^{n}{\left(1-\mu_{{\overset{ˇ}{c}}_{ij}}\right)}^{{Ω}_{i}}\right)}^{\gamma_{j}}\leq {\left(1-\begin{array}{c} min\\ j \end{array}\begin{array}{c} min\\ i \end{array}\left\{\mu_{{\overset{ˇ}{c}}_{ij}}\right\}\right)}^{\sum_{j=1}^{m}\gamma_{j}}\\ \Leftrightarrow 1-\begin{array}{c} max\\ j \end{array}\begin{array}{c} max\\ i \end{array}\left\{\mu_{{\overset{ˇ}{\mu }}_{ij}}\right\}\leq \prod_{j=1}^{m}{\left(\prod_{i=1}^{n}{\left(1-\mu_{{\overset{ˇ}{c}}_{ij}}\right)}^{{Ω}_{i}}\right)}^{\gamma_{j}}\leq 1-\begin{array}{c} min\\ j \end{array}\begin{array}{c} min\\ i \end{array}\left\{\mu_{{\overset{ˇ}{c}}_{ij}}\right\}\\ \Leftrightarrow \begin{array}{c} min\\ j \end{array}\begin{array}{c} min\\ i \end{array}\left\{\mu_{{\overset{ˇ}{c}}_{ij}}\right\}\leq 1-\prod_{j=1}^{m}{\left(\prod_{i=1}^{n}{\left(1-\mu_{{\overset{ˇ}{c}}_{ij}}\right)}^{{Ω}_{i}}\right)}^{\gamma_{j}}\leq \begin{array}{c} max\\ j \end{array}\begin{array}{c} max\\ i \end{array}\left\{\mu_{{\overset{ˇ}{c}}_{ij}}\right\}\\ \Leftrightarrow \begin{array}{c} min\\ j \end{array}\begin{array}{c} min\\ i \end{array}\left\{\mu_{{\overset{ˇ}{c}}_{ij}}\right\}\leq 1-\prod_{j=1}^{m}{\left(\prod_{i=1}^{n}{\left(1-\mu_{{\overset{ˇ}{c}}_{ij}}\right)}^{{Ω}_{i}}\right)}^{\gamma_{j}}\leq \begin{array}{c} max\\ j \end{array}\begin{array}{c} max\\ i \end{array}\left\{\mu_{{\overset{ˇ}{c}}_{ij}}\right\} \end{array}$$Similarly, (6)$$\begin{array}{c} min\\ j \end{array}\begin{array}{c} min\\ i \end{array}\left\{\vartheta_{{\overset{ˇ}{c}}_{ij}}\right\}\leq \prod_{j=1}^{m}{\left(\prod_{i=1}^{n}{\left(\vartheta_{{\overset{ˇ}{c}}_{ij}}\right)}^{{Ω}_{i}}\right)}^{\gamma_{j}}\leq \begin{array}{c} max\\ j \end{array}\begin{array}{c} max\\ i \end{array}\left\{\vartheta_{{\overset{ˇ}{c}}_{ij}}\right\}$$Let $IFHSWA\;\left(J_{{\overset{ˇ}{c}}_{11}},\;J_{{\overset{ˇ}{c}}_{12}},\ldots ,\;J_{{\overset{ˇ}{c}}_{nm}}\right)$ = $\mu_{{\overset{ˇ}{c}}_{\delta }},\;\vartheta_{{\overset{ˇ}{c}}_{\delta }}$ = $J_{{\overset{ˇ}{c}}_{\delta }}$, that inequities 5 and 6 could be turned into the subsequent form:$\begin{array}{c} min\\ j \end{array}\begin{array}{c} min\\ i \end{array}\left\{\mu_{{\overset{ˇ}{c}}_{ij}}\right\}\leq$ $\mu_{{\overset{ˇ}{c}}_{\delta }}$ ≤ $\begin{array}{c} max\\ j \end{array}\begin{array}{c} max\\ i \end{array}\left\{\mu_{{\overset{ˇ}{c}}_{ij}}\right\}$ and $\begin{array}{c} min\\ j \end{array}\begin{array}{c} min\\ i \end{array}\left\{\vartheta_{{\overset{ˇ}{c}}_{ij}}\right\}$ ≤ $\vartheta_{{\overset{ˇ}{c}}_{\delta }}$ ≤ $\begin{array}{c} max\\ j \end{array}\begin{array}{c} max\\ i \end{array}\left\{\vartheta_{{\overset{ˇ}{c}}_{ij}}\right\}$ respectively.So, by utilizing Equation (1), we have:$S\left(J_{{\overset{ˇ}{c}}_{\delta }}\right)$ = $\mu_{{\overset{ˇ}{c}}_{\delta }}-\vartheta_{{\overset{ˇ}{c}}_{\delta }}$ ≤ $\begin{array}{c} max\\ j \end{array}\begin{array}{c} max\\ i \end{array}\left\{\mu_{{\overset{ˇ}{c}}_{ij}}\right\}-$ $\begin{array}{c} min\\ j \end{array}\begin{array}{c} min\\ i \end{array}\left\{\vartheta_{{\overset{ˇ}{c}}_{ij}}\right\}$ = $S\left(J_{{\overset{ˇ}{c}}_{ij}}{}^{+}\right)$,$S\left(J_{{\overset{ˇ}{c}}_{\delta }}\right)$ = $\mu_{{\overset{ˇ}{c}}_{\delta }}-\vartheta_{{\overset{ˇ}{c}}_{\delta }}$ ≥ $\begin{array}{c} min\\ j \end{array}\begin{array}{c} min\\ i \end{array}\left\{\mu_{{\overset{ˇ}{c}}_{\delta }}\right\}-$ $\begin{array}{c} max\\ j \end{array}\begin{array}{c} max\\ i \end{array}\left\{\vartheta_{{\overset{ˇ}{c}}_{\delta }}\right\}$ = $S\left(J_{{\overset{ˇ}{c}}_{ij}}{}^{-}\right)$. Then, by order relation among two IFHSNs, we have$J_{{\overset{ˇ}{c}}_{ij}}{}^{-}$ ≤ $IFHSWA\;\left(J_{{\overset{ˇ}{c}}_{11}},\;J_{{\overset{ˇ}{c}}_{12}},\ldots ,\;J_{{\overset{ˇ}{c}}_{nm}}\right)$ ≤ $J_{{\overset{ˇ}{c}}_{ij}}{}^{+}$ □

#### 3.2.3. (Shift Invariance) If $J_{{\overset{ˇ}{c}}_{\delta }}$ = $\left\langle \mu_{{\overset{ˇ}{c}}_{\delta }},\vartheta_{{\overset{ˇ}{c}}_{\delta }}\right\rangle$ Be an IFHSN. Then

$IFHSWA\;\left(J_{c_{11}}⊕J_{{\overset{ˇ}{c}}_{\delta }},\;J_{{\overset{ˇ}{c}}_{12}}⊕J_{{\overset{ˇ}{c}}_{\delta }},\ldots ,\;J_{{\overset{ˇ}{c}}_{nm}}⊕J_{{\overset{ˇ}{c}}_{\delta }}\right)$ = $IFHSWA\;\left(J_{{\overset{ˇ}{c}}_{11}},\;J_{{\overset{ˇ}{c}}_{12}},\ldots ,\;J_{{\overset{ˇ}{c}}_{nm}}\right)⊕J_{{\overset{ˇ}{c}}_{\delta }}$.

**Proof.** Consider $J_{{\overset{ˇ}{c}}_{\delta }}$ and $J_{{\overset{ˇ}{c}}_{ij}}$ to be two IFHSNs. Then, by using operational laws defined under IFHSNs in Definition 5 (1), we have:$J_{{\overset{ˇ}{c}}_{\delta }}⊕J_{c_{ij}}$ = $\left\langle \mu_{{\overset{ˇ}{c}}_{\delta }}+\mu_{{\overset{ˇ}{c}}_{ij}}-\mu_{{\overset{ˇ}{c}}_{\delta }}\mu_{{\overset{ˇ}{c}}_{ij}},\;\vartheta_{{\overset{ˇ}{c}}_{\delta }}\vartheta_{{\overset{ˇ}{c}}_{ij}}\right\rangle$, therefore
$$\begin{array}{l} IFHSWA\;\left(J_{{\overset{ˇ}{c}}_{11}}⊕J_{{\overset{ˇ}{c}}_{\delta }},\;J_{{\overset{ˇ}{c}}_{12}}⊕J_{{\overset{ˇ}{c}}_{\delta }},\ldots ,\;J_{{\overset{ˇ}{c}}_{nm}}⊕J_{{\overset{ˇ}{c}}_{\delta }}\right)={⊕}_{\;j=1}^{m}\gamma_{j}\left({⊕}_{i=1}^{n}{Ω}_{i}\left(J_{{\overset{ˇ}{c}}_{ij}}⊕J_{{\overset{ˇ}{c}}_{\delta }}\right)\;\right)\\ =\langle 1-\prod_{j=1}^{m}{\left(\prod_{i=1}^{n}{\left(1-\mu_{{\overset{ˇ}{c}}_{ij}}\right)}^{{Ω}_{i}}{\left(1-\mu_{{\overset{ˇ}{c}}_{\delta }}\right)}^{{Ω}_{i}}\right)}^{\gamma_{j}},\;\prod_{j=1}^{m}{\left(\prod_{i=1}^{n}{\left(\vartheta_{{\overset{ˇ}{c}}_{ij}}\right)}^{{Ω}_{i}}{\left(\vartheta_{{\overset{ˇ}{c}}_{\delta }}\right)}^{{Ω}_{i}}\right)}^{\gamma_{j}}\rangle \\ =\langle 1-\left(1-\mu_{{\overset{ˇ}{c}}_{\delta }}\right)\prod_{j=1}^{m}{\left(\prod_{i=1}^{n}{\left(1-\mu_{{\overset{ˇ}{c}}_{ij}}\right)}^{{Ω}_{i}}\right)}^{\gamma_{j}},\vartheta_{{\overset{ˇ}{c}}_{\delta }}\prod_{j=1}^{m}{\left(\prod_{i=1}^{n}{\left(\vartheta_{{\overset{ˇ}{c}}_{ij}}\right)}^{{Ω}_{i}}\right)}^{\gamma_{j}}\rangle \\ =\langle 1-\prod_{j=1}^{m}{\left(\prod_{i=1}^{n}{\left(1-\mu_{{\overset{ˇ}{c}}_{ij}}\right)}^{{Ω}_{i}}\right)}^{\gamma_{j}},\;\prod_{j=1}^{m}{\left(\prod_{i=1}^{n}{\left(\vartheta_{{\overset{ˇ}{c}}_{ij}}\right)}^{{Ω}_{i}}\right)}^{\gamma_{j}}\rangle ⊕\left\langle \mu_{{\overset{ˇ}{c}}_{\delta }},\;\vartheta_{{\overset{ˇ}{c}}_{\delta }}\right\rangle \\ =IFHSWA\;\left(J_{c_{11}},\;J_{{\overset{ˇ}{c}}_{12}},\ldots ,\;J_{{\overset{ˇ}{c}}_{nm}}\right)⊕J_{{\overset{ˇ}{c}}_{\delta }}. \end{array}$$Which completes the proof □

#### 3.2.4. (Homogeneity) 

Prove that $IFHSWA\;\left(\alpha J_{{\overset{ˇ}{c}}_{11}},\;\alpha J_{{\overset{ˇ}{c}}_{12}},\ldots ,\;\alpha J_{{\overset{ˇ}{c}}_{nm}}\right)$ = α $IFHSWA\;\left(J_{{\overset{ˇ}{c}}_{11}},\;J_{{\overset{ˇ}{c}}_{12}},\ldots ,\;J_{{\overset{ˇ}{c}}_{nm}}\right)$ for any positive real number α.

**Proof.** Let $J_{{\overset{ˇ}{c}}_{ij}}$ be an IFHSN and >0, then by using Definition 5 (3), we have$\alpha J_{{\overset{ˇ}{c}}_{ij}}$ = $\left\langle 1-{\left(1-\mu_{{\overset{ˇ}{c}}_{ij}}\right)}^{\alpha },\;\vartheta_{{\overset{ˇ}{c}}_{ij}}{}^{\alpha }\right\rangle$. So,
$$\begin{array}{l} IFHSWA\;\left(\alpha J_{{\overset{ˇ}{c}}_{11}},\;\alpha J_{{\overset{ˇ}{c}}_{12}},\ldots ,\;\alpha J_{{\overset{ˇ}{c}}_{nm}}\right)\\ =\langle 1-\prod_{j=1}^{m}{\left(\prod_{i=1}^{n}{\left(1-\mu_{{\overset{ˇ}{c}}_{ij}}\right)}^{\alpha {Ω}_{i}}\right)}^{\gamma_{j}},\;\prod_{j=1}^{m}{\left(\prod_{i=1}^{n}{\left(\vartheta_{{\overset{ˇ}{c}}_{ij}}\right)}^{\alpha {Ω}_{i}}\right)}^{\gamma_{j}}\rangle \\ =\langle 1-{\left(\prod_{j=1}^{m}{\left(\prod_{i=1}^{n}{\left(1-\mu_{{\overset{ˇ}{c}}_{ij}}\right)}^{{Ω}_{i}}\right)}^{\gamma_{j}}\right)}^{\alpha },\;{\left(\prod_{j=1}^{m}{\left(\prod_{i=1}^{n}{\left(\vartheta_{{\overset{ˇ}{c}}_{ij}}\right)}^{{Ω}_{i}}\right)}^{\gamma_{j}}\right)}^{\alpha }\rangle \\ =\alpha \;IFHSWA\;\left(J_{{\overset{ˇ}{c}}_{11}},\;J_{{\overset{ˇ}{c}}_{12}},\ldots ,\;J_{{\overset{ˇ}{c}}_{nm}}\right). \end{array}$$Completes the proof. □

**Definition** **7.***Let*$J_{{\overset{ˇ}{c}}_{k}}$*=*$\left(\mu_{{\overset{ˇ}{c}}_{k}},\;\vartheta_{{\overset{ˇ}{c}}_{k}}\right)$*be an IFHSN,*${Ω}_{i}$*and*$\gamma_{j}$*be weight vectors for expert’s and multi sub-attributes of the considered attributes correspondingly along with specified circumstances*${Ω}_{i}$ >*0,*$\sum_{i=1}^{n}{Ω}_{i}$*= 1,*$\gamma_{j}$ >*0,*$\sum_{\;j=1}^{m}\gamma_{j}$*= 1. Then IFHSWG operator can be defined as follows:*IFHSWG: $\Delta^{n}$ → Δ
*defined as follows:*(7)$$IFHSWG\;\left(J_{{\overset{ˇ}{c}}_{11}},\;J_{{\overset{ˇ}{c}}_{12}},\ldots ,\;J_{{\overset{ˇ}{c}}_{nm}}\right)={⊗}_{j=1}^{m}{\left({⊗}_{i=1}^{n}J_{{\overset{ˇ}{c}}_{nm}}^{{Ω}_{i}}\right)}^{\gamma_{j}}.$$

**Theorem** **2.***Let*$J_{{\overset{ˇ}{c}}_{k}}$*=*$\left(\mu_{{\overset{ˇ}{c}}_{k}},\;\vartheta_{{\overset{ˇ}{c}}_{k}}\right)$*be an IFHSN. Then, utilizing Equation (7), we get PFHSN and*(8)$$IFHSWG\;\left(J_{{\overset{ˇ}{c}}_{11}},\;J_{{\overset{ˇ}{c}}_{12}},\ldots ,\;J_{{\overset{ˇ}{c}}_{nm}}\right)=\;\left\langle \prod_{j=1}^{m}{\left(\prod_{i=1}^{n}{\left(\mu_{{\overset{ˇ}{c}}_{ij}}\right)}^{{Ω}_{i}}\right)}^{\gamma_{j}},\;1-\prod_{j=1}^{m}{\left(\prod_{i=1}^{n}{\left(1-\vartheta_{{\overset{ˇ}{c}}_{ij}}\right)}^{{Ω}_{i}}\right)}^{\gamma_{j}}\right\rangle$$${Ω}_{i}$*and*$\gamma_{j}$*be weight vectors for experts and multi sub-attributes of the considered attributes correspondingly along with specified circumstances*${Ω}_{i}$ >*0,*$\sum_{i=1}^{n}{Ω}_{i}$*= 1,*$\gamma_{j}$ >*0,*$\sum_{\;j=1}^{m}\gamma_{j}$*= 1.*

**Proof.** The IFHSWA can be proven as follows utilizing the principle of mathematical induction.For n=1, we get ${Ω}_{1}$ = 1. Then, we have
$$\begin{array}{l} IFHSWG\;\left(J_{{\overset{ˇ}{c}}_{11}},\;J_{{\overset{ˇ}{c}}_{12}},\ldots ,\;J_{{\overset{ˇ}{c}}_{nm}}\right)={⊗}_{j=1}^{m}J_{{\overset{ˇ}{c}}_{1j}}^{\gamma_{j}}\\ IFHSWG\;\left(J_{{\overset{ˇ}{c}}_{11}},\;J_{{\overset{ˇ}{c}}_{12}},\ldots ,\;J_{{\overset{ˇ}{c}}_{nm}}\right)=\langle \prod_{j=1}^{m}{\left(\mu_{{\overset{ˇ}{c}}_{1j}}\right)}^{\gamma_{j}},\;1-\prod_{j=1}^{m}{\left(1-\vartheta_{{\overset{ˇ}{c}}_{1j}}\right)}^{\gamma_{j}}\rangle \\ =\langle \prod_{j=1}^{m}{\left(\prod_{i=1}^{1}{\left(\mu_{{\overset{ˇ}{c}}_{ij}}\right)}^{{Ω}_{i}}\right)}^{\gamma_{j}},\;1-\prod_{j=1}^{m}{\left(\prod_{i=1}^{1}{\left(1-\vartheta_{{\overset{ˇ}{c}}_{1j}}\right)}^{{Ω}_{i}}\right)}^{\gamma_{j}}\rangle . \end{array}$$For m=1, we get $\gamma_{1}$ = 1. Then, we have
$$\begin{array}{l} IFHSWG\;\left(J_{{\overset{ˇ}{c}}_{11}},\;J_{{\overset{ˇ}{c}}_{12}},\ldots ,\;J_{{\overset{ˇ}{c}}_{nm}}\right)={⊗}_{i=1}^{n}{\left(J_{{\overset{ˇ}{c}}_{i1}}\right)}^{{Ω}_{i}}\\ =\langle \prod_{i=1}^{n}{\left(\mu_{{\overset{ˇ}{c}}_{i1}}\right)}^{{Ω}_{i}},\;1-\prod_{i=1}^{n}{\left(1-\vartheta_{{\overset{ˇ}{c}}_{i1}}\right)}^{{Ω}_{i}}\rangle \\ =\langle \prod_{j=1}^{1}{\left(\prod_{i=1}^{n}{\left(\mu_{{\overset{ˇ}{c}}_{ij}}\right)}^{{Ω}_{i}}\right)}^{\gamma_{j}},\;1-\prod_{j=1}^{1}{\left(\prod_{i=1}^{n}{\left(1-\vartheta_{{\overset{ˇ}{c}}_{ij}}\right)}^{{Ω}_{i}}\right)}^{\gamma_{j}}\rangle . \end{array}$$This shows that Equation (8) fulfills for n=1 and m=1. Consider Equation (8) holds for $m=\beta_{1}+1$, $n=\beta_{2}$ and $m=\beta_{1}$, $n=\beta_{2}+1$, such as: $$\begin{array}{l} {⊗}_{\;j=1}^{\beta_{1}+1}{\left({⊗}_{i=1}^{\beta_{2}}{\left(J_{{\overset{ˇ}{c}}_{ij}}\right)}^{{Ω}_{i}}\right)}^{\gamma_{j}}=\langle \prod_{j=1}^{\beta_{1}+1}{\left(\prod_{i=1}^{\beta_{2}}{\left(\mu_{{\overset{ˇ}{c}}_{ij}}\right)}^{{Ω}_{i}}\right)}^{\gamma_{j}},1-\prod_{j=1}^{\beta_{1}+1}{\left(\prod_{i=1}^{\beta_{2}}{\left(1-\vartheta_{{\overset{ˇ}{c}}_{ij}}\right)}^{{Ω}_{i}}\right)}^{\gamma_{j}}\rangle \\ {⊗}_{\;j=1}^{\beta_{1}}{\left({⊗}_{i=1}^{\beta_{2}}{\left(J_{{\overset{ˇ}{c}}_{ij}}\right)}^{{Ω}_{i}}\right)}^{\gamma_{j}}=\langle \prod_{j=1}^{\beta_{1}}{\left(\prod_{i=1}^{\beta_{2}+1}{\left(\mu_{{\overset{ˇ}{c}}_{ij}}\right)}^{{Ω}_{i}}\right)}^{\gamma_{j}},1-\prod_{j=1}^{\beta_{1}}{\left(\prod_{i=1}^{\beta_{2}+1}{\left(1-\vartheta_{{\overset{ˇ}{c}}_{ij}}\right)}^{{Ω}_{i}}\right)}^{\gamma_{j}}\rangle \end{array}$$For $m=\beta_{1}+1$ and $n=\beta_{2}+1$, we have
$$\begin{array}{l} {⊗}_{\;j=1}^{\beta_{1}+1}{\left({⊗}_{i=1}^{\beta_{2}+1}{\left(J_{{\overset{ˇ}{c}}_{ij}}\right)}^{{Ω}_{i}}\right)}^{\gamma_{j}}={⊗}_{\;j=1}^{\beta_{1}+1}{\left({⊗}_{i=1}^{\beta_{2}}{\left(J_{{\overset{ˇ}{c}}_{ij}}\right)}^{{Ω}_{i}}⊗{\left(J_{{\overset{ˇ}{c}}_{\left(\beta_{2}+1\right)j}}\right)}^{{Ω}_{\beta_{2}+1}}\;\right)}^{\gamma_{j}}\\ ={⊗}_{\;j=1}^{\beta_{1}+1}{\left({⊗}_{i=1}^{\beta_{2}}{\left(J_{{\overset{ˇ}{c}}_{ij}}\right)}^{{Ω}_{i}}\;\right)}^{\gamma_{j}}{⊗}_{\;j=1}^{\beta_{1}+1}{\left({\left(J_{{\overset{ˇ}{c}}_{\left(\beta_{2}+1\right)j}}\right)}^{{Ω}_{\beta_{2}+1}}\;\right)}^{\gamma_{j}}\\ =\langle \begin{array}{c} \prod_{j=1}^{\beta_{1}+1}{\left(\prod_{i=1}^{\beta_{2}}{\left(\mu_{{\overset{ˇ}{c}}_{ij}}\right)}^{{Ω}_{i}}\right)}^{\gamma_{j}}⊗\prod_{j=1}^{\beta_{1}+1}{\left({\left(\mu_{{\overset{ˇ}{c}}_{\left(\beta_{2}+1\right)j}}\right)}^{{Ω}_{\beta_{2}+1}}\right)}^{\gamma_{j}},\\ 1-\prod_{j=1}^{\beta_{1}+1}{\left(\prod_{i=1}^{\beta_{2}}{\left(1-\vartheta_{{\overset{ˇ}{c}}_{ij}}\right)}^{{Ω}_{i}}\right)}^{\gamma_{j}}⊗1-\prod_{j=1}^{\beta_{1}+1}{\left({\left(1-\vartheta_{{\overset{ˇ}{c}}_{\left(\beta_{2}+1\right)j}}\right)}^{{Ω}_{\beta_{2}+1}}\right)}^{\gamma_{j}} \end{array}\rangle \\ =\prod_{j=1}^{\beta_{1}+1}{\left(\prod_{i=1}^{\beta_{2}+1}{\left(\mu_{{\overset{ˇ}{c}}_{ij}}\right)}^{{Ω}_{i}}\right)}^{\gamma_{j}},\;1-\prod_{j=1}^{\beta_{1}+1}{\left(\prod_{i=1}^{\beta_{2}+1}{\left(1-\vartheta_{{\overset{ˇ}{c}}_{ij}}\right)}^{{Ω}_{i}}\right)}^{\gamma_{j}}. \end{array}$$Hence, it is true for $m=\beta_{1}+1$ and $n=\beta_{2}+1$.  □

We establish some properties for the collection of IFHSNs based on Theorem 2, by utilizing the proposed IFHSWG operator.

### 3.3. Properties of IFHSWG Operator 

#### 3.3.1. (Idempotency) 

$J_{{\overset{ˇ}{c}}_{ij}}$ = $J_{{\overset{ˇ}{c}}_{\delta }}$ = $\left(\mu_{{\overset{ˇ}{c}}_{ij}},\;\vartheta_{{\overset{ˇ}{c}}_{ij}}\right)$ ∀ i, j, then, 

$IFHSWG\;\left(J_{{\overset{ˇ}{c}}_{11}},\;J_{{\overset{ˇ}{c}}_{12}},\ldots ,\;J_{{\overset{ˇ}{c}}_{nm}}\right)$ = $J_{{\overset{ˇ}{c}}_{\delta }}$.

**Proof.** As we know $J_{{\overset{ˇ}{c}}_{ij}}$ = $J_{{\overset{ˇ}{c}}_{\delta }}$ = $\left(\mu_{{\overset{ˇ}{c}}_{ij}},\;\vartheta_{{\overset{ˇ}{c}}_{ij}}\right)$ to be a collection of IFHSNs, then by Equation (8)
$$\begin{array}{l} IFHSWG\;\;\left(J_{{\overset{ˇ}{c}}_{11}},\;J_{{\overset{ˇ}{c}}_{12}},\ldots ,\;J_{{\overset{ˇ}{c}}_{nm}}\right)=\langle \prod_{j=1}^{m}{\left(\prod_{i=1}^{n}{\left(\mu_{{\overset{ˇ}{c}}_{ij}}\right)}^{{Ω}_{i}}\right)}^{\gamma_{j}},1-\prod_{j=1}^{m}{\left(\prod_{i=1}^{n}{\left(1-\vartheta_{{\overset{ˇ}{c}}_{ij}}\right)}^{{Ω}_{i}}\right)}^{\gamma_{j}}\rangle \\ =\langle {\left({\left(\mu_{{\overset{ˇ}{c}}_{ij}}\right)}^{\sum_{i=1}^{n}{Ω}_{i}}\right)}^{\sum_{j=1}^{m}\gamma_{j}},\;1-{\left({\left(1-\vartheta_{{\overset{ˇ}{c}}_{ij}}\right)}^{\sum_{i=1}^{n}{Ω}_{i}}\right)}^{\sum_{j=1}^{m}\gamma_{j}}\rangle \\ =\langle \mu_{{\overset{ˇ}{c}}_{ij}},\;1-\left(1-\vartheta_{{\overset{ˇ}{c}}_{ij}})\right\rangle =\left(\mu_{{\overset{ˇ}{c}}_{ij}},\;\vartheta_{{\overset{ˇ}{c}}_{ij}}\right)=J_{{\overset{ˇ}{c}}_{\delta }}.\;□ \end{array}$$

#### 3.3.2. (Boundedness) 

Let $J_{{\overset{ˇ}{c}}_{ij}}$ be a collection of IFHSNs and

Jcˇij− = ⟨minjmini{μcˇij}, maxjmaxi{ϑcˇij}⟩ and Jcˇij+ = ⟨maxjmaxi{μcˇij}, minjmini{ϑcˇij}⟩, then

$J_{{\overset{ˇ}{c}}_{ij}}{}^{-}$ ≤ $IFHSWG\;\left(J_{{\overset{ˇ}{c}}_{11}},\;J_{{\overset{ˇ}{c}}_{12}},\ldots ,\;J_{{\overset{ˇ}{c}}_{nm}}\right)$ ≤ $J_{{\overset{ˇ}{c}}_{ij}}{}^{+}$.

**Proof.** As we know that $J_{{\overset{ˇ}{c}}_{ij}}$ = $\left(\mu_{{\overset{ˇ}{c}}_{ij}},\;J_{{\overset{ˇ}{c}}_{ij}}\right)$ is an IFHSN, then
(9)$$\begin{array}{l} \begin{array}{c} min\\ j \end{array}\begin{array}{c} min\\ i \end{array}\left\{\mu_{{\overset{ˇ}{c}}_{ij}}\right\}\leq \mu_{{\overset{ˇ}{c}}_{ij}}\leq \begin{array}{c} max\\ j \end{array}\begin{array}{c} max\\ i \end{array}\left\{\mu_{{\overset{ˇ}{c}}_{ij}}\right\}\\ \Rightarrow \begin{array}{c} min\\ j \end{array}\begin{array}{c} min\\ i \end{array}\left\{\mu_{{\overset{ˇ}{c}}_{ij}}\right\}\leq \mu_{{\overset{ˇ}{c}}_{ij}}\leq \begin{array}{c} max\\ j \end{array}\begin{array}{c} max\\ i \end{array}\left\{\mu_{{\overset{ˇ}{c}}_{ij}}\right\}\\ \Leftrightarrow {\left(\begin{array}{c} min\\ j \end{array}\begin{array}{c} min\\ i \end{array}\left\{\mu_{{\overset{ˇ}{c}}_{ij}}\right\}\right)}^{{Ω}_{i}}\leq {\left(\mu_{{\overset{ˇ}{c}}_{ij}}\right)}^{{Ω}_{i}}\leq {\left(\begin{array}{c} max\\ j \end{array}\begin{array}{c} max\\ i \end{array}\left\{\mu_{{\overset{ˇ}{c}}_{ij}}\right\}\right)}^{{Ω}_{i}}\\ \Leftrightarrow {\left(\begin{array}{c} min\\ j \end{array}\begin{array}{c} min\\ i \end{array}\left\{\mu_{{\overset{ˇ}{c}}_{ij}}\right\}\right)}^{\sum_{i=1}^{n}{Ω}_{i}}\leq \prod_{i=1}^{n}{\left(\mu_{{\overset{ˇ}{c}}_{ij}}\right)}^{{Ω}_{i}}\leq {\left(\begin{array}{c} max\\ j \end{array}\begin{array}{c} max\\ i \end{array}\left\{\mu_{{\overset{ˇ}{c}}_{ij}}\right\}\right)}^{\sum_{i=1}^{n}{Ω}_{i}}\\ \Leftrightarrow {\left(\begin{array}{c} min\\ j \end{array}\begin{array}{c} min\\ i \end{array}\left\{\mu_{{\overset{ˇ}{c}}_{ij}}\right\}\right)}^{\sum_{j=1}^{m}\gamma_{j}}\leq \prod_{j=1}^{m}{\left(\prod_{i=1}^{n}{\left(\mu_{{\overset{ˇ}{c}}_{ij}}\right)}^{{Ω}_{i}}\right)}^{\gamma_{j}}\leq {\left(\begin{array}{c} max\\ j \end{array}\begin{array}{c} max\\ i \end{array}\left\{\mu_{{\overset{ˇ}{c}}_{ij}}\right\}\right)}^{\sum_{j=1}^{m}\gamma_{j}}\\ \Leftrightarrow \begin{array}{c} min\\ j \end{array}\begin{array}{c} min\\ i \end{array}\left\{\mu_{{\overset{ˇ}{c}}_{ij}}\right\}\leq \prod_{j=1}^{m}{\left(\prod_{i=1}^{n}{\left(\mu_{{\overset{ˇ}{c}}_{ij}}\right)}^{{Ω}_{i}}\right)}^{\gamma_{j}}\leq \begin{array}{c} max\\ j \end{array}\begin{array}{c} max\\ i \end{array}\left\{\mu_{{\overset{ˇ}{c}}_{ij}}\right\}\\ \Leftrightarrow \begin{array}{c} min\\ j \end{array}\begin{array}{c} min\\ i \end{array}\left\{\mu_{{\overset{ˇ}{c}}_{ij}}\right\}\leq \prod_{j=1}^{m}{\left(\prod_{i=1}^{n}{\left(\mu_{{\overset{ˇ}{c}}_{ij}}\right)}^{{Ω}_{i}}\right)}^{\gamma_{j}}\leq \begin{array}{c} max\\ j \end{array}\begin{array}{c} max\\ i \end{array}\left\{\mu_{{\overset{ˇ}{c}}_{ij}}\right\}\\ \Leftrightarrow \begin{array}{c} min\\ j \end{array}\begin{array}{c} min\\ i \end{array}\left\{\mu_{{\overset{ˇ}{c}}_{ij}}\right\}\leq \prod_{j=1}^{m}{\left(\prod_{i=1}^{n}{\left(\mu_{{\overset{ˇ}{c}}_{ij}}\right)}^{{Ω}_{i}}\right)}^{\gamma_{j}}\leq \begin{array}{c} max\\ j \end{array}\begin{array}{c} max\\ i \end{array}\left\{\mu_{{\overset{ˇ}{c}}_{ij}}\right\} \end{array}$$Similarly, (10)$$\Leftrightarrow \begin{array}{c} min\\ j \end{array}\begin{array}{c} min\\ i \end{array}\left\{\vartheta_{{\overset{ˇ}{c}}_{ij}}\right\}\leq 1-\prod_{j=1}^{m}{\left(\prod_{i=1}^{n}{\left(1-\vartheta_{{\overset{ˇ}{c}}_{ij}}\right)}^{{Ω}_{i}}\right)}^{\gamma_{j}}\leq \begin{array}{c} max\\ j \end{array}\begin{array}{c} max\\ i \end{array}\left\{\vartheta_{{\overset{ˇ}{c}}_{ij}}\right\}$$Let $IFHSWG\;\left(J_{{\overset{ˇ}{c}}_{11}},\;J_{{\overset{ˇ}{c}}_{12}},\ldots ,\;J_{{\overset{ˇ}{c}}_{nm}}\right)$ = $\left\langle \mu_{{\overset{ˇ}{c}}_{\delta }},\;\vartheta_{{\overset{ˇ}{c}}_{\delta }}\right\rangle$ = $J_{{\overset{ˇ}{c}}_{\delta }}$, then Inequalities 9 and 10 can be transformed into the following form:$\begin{array}{c} min\\ j \end{array}\begin{array}{c} min\\ i \end{array}\left\{\mu_{{\overset{ˇ}{c}}_{ij}}\right\}\leq$ $\mu_{{\overset{ˇ}{c}}_{\delta }}$ ≤ $\begin{array}{c} max\\ j \end{array}\begin{array}{c} max\\ i \end{array}\left\{\mu_{{\overset{ˇ}{c}}_{ij}}\right\}$ and $\begin{array}{c} min\\ j \end{array}\begin{array}{c} min\\ i \end{array}\left\{\vartheta_{{\overset{ˇ}{c}}_{ij}}\right\}$ ≤ $\vartheta_{{\overset{ˇ}{c}}_{\delta }}$ ≤ $\begin{array}{c} max\\ j \end{array}\begin{array}{c} max\\ i \end{array}\left\{\vartheta_{{\overset{ˇ}{c}}_{ij}}\right\}$ respectively.So, by utilizing Equation (1), we have:$S\left(J_{{\overset{ˇ}{c}}_{\delta }}\right)$ = $\mu_{{\overset{ˇ}{c}}_{\delta }}-\vartheta_{{\overset{ˇ}{c}}_{\delta }}$ ≤ $\begin{array}{c} max\\ j \end{array}\begin{array}{c} max\\ i \end{array}\left\{\mu_{{\overset{ˇ}{c}}_{ij}}\right\}-$ $\begin{array}{c} min\\ j \end{array}\begin{array}{c} min\\ i \end{array}\left\{\vartheta_{{\overset{ˇ}{c}}_{ij}}\right\}$ = $S\left(J_{{\overset{ˇ}{c}}_{ij}}{}^{+}\right)$,$S\left(J_{{\overset{ˇ}{c}}_{\delta }}\right)$ = $\mu_{{\overset{ˇ}{c}}_{\delta }}-\vartheta_{{\overset{ˇ}{c}}_{\delta }}$ ≥ $\begin{array}{c} min\\ j \end{array}\begin{array}{c} min\\ i \end{array}\left\{\mu_{{\overset{ˇ}{c}}_{\delta }}\right\}-$ $\begin{array}{c} max\\ j \end{array}\begin{array}{c} max\\ i \end{array}\left\{\vartheta_{{\overset{ˇ}{c}}_{\delta }}\right\}$ = $S\left(J_{{\overset{ˇ}{c}}_{ij}}{}^{-}\right)$. Then, by order relation among two IFHSNs, we have$J_{{\overset{ˇ}{c}}_{ij}}{}^{-}$ ≤ $IFHSWG\;\left(J_{{\overset{ˇ}{c}}_{11}},\;J_{{\overset{ˇ}{c}}_{12}},\ldots ,\;J_{{\overset{ˇ}{c}}_{nm}}\right)$ ≤ $J_{{\overset{ˇ}{c}}_{ij}}{}^{+}$.  □

#### 3.3.3. (Shift Invariance) 

If $J_{{\overset{ˇ}{c}}_{\delta }}$ = $\left\langle \mu_{{\overset{ˇ}{c}}_{\delta }},\;\vartheta_{{\overset{ˇ}{c}}_{\delta }}\right\rangle$ be an IFHSN. Then, 

$IFHSWG\;\left(J_{{\overset{ˇ}{c}}_{11}}⊗J_{{\overset{ˇ}{c}}_{\delta }},\;J_{{\overset{ˇ}{c}}_{12}}⊗J_{{\overset{ˇ}{c}}_{\delta }},\ldots ,\;J_{{\overset{ˇ}{c}}_{nm}}⊗J_{{\overset{ˇ}{c}}_{\delta }}\right)$ = $IFHSWG\;\left(J_{{\overset{ˇ}{c}}_{11}},\;J_{{\overset{ˇ}{c}}_{12}},\ldots ,\;J_{{\overset{ˇ}{c}}_{nm}}\right)⊗J_{{\overset{ˇ}{c}}_{\delta }}$.

**Proof.** Let $J_{{\overset{ˇ}{c}}_{\delta }}$ and $J_{{\overset{ˇ}{c}}_{ij}}$ be two IFHSNs. Utilizing Definition 5 (2), we have:$J_{{\overset{ˇ}{c}}_{\delta }}⊗J_{c_{ij}}$ = $\left\langle \mu_{{\overset{ˇ}{c}}_{11}}\mu_{{\overset{ˇ}{c}}_{12}},\;\vartheta_{{\overset{ˇ}{c}}_{11}}+\vartheta_{{\overset{ˇ}{c}}_{12}}-\vartheta_{{\overset{ˇ}{c}}_{11}}\vartheta_{{\overset{ˇ}{c}}_{12}}\right\rangle$. Therefore
$$\begin{array}{l} IFHSWG\;\left(J_{{\overset{ˇ}{c}}_{11}}⊗J_{{\overset{ˇ}{c}}_{\delta }},\;J_{{\overset{ˇ}{c}}_{12}}⊗J_{{\overset{ˇ}{c}}_{\delta }},\ldots ,\;J_{{\overset{ˇ}{c}}_{nm}}⊗J_{{\overset{ˇ}{c}}_{\delta }}\right)={⊗}_{\;j=1}^{m}\gamma_{j}\left({⊗}_{i=1}^{n}{Ω}_{i}\left(J_{{\overset{ˇ}{c}}_{ij}}⊗J_{{\overset{ˇ}{c}}_{\delta }}\right)\;\right)\\ =\langle \prod_{j=1}^{m}{\left(\prod_{i=1}^{n}{\left(\mu_{{\overset{ˇ}{c}}_{ij}}\right)}^{{Ω}_{i}}{\left(\mu_{{\overset{ˇ}{c}}_{\delta }}\right)}^{{Ω}_{i}}\right)}^{\gamma_{j}},\;1-\prod_{j=1}^{m}{\left(\prod_{i=1}^{n}{\left(1-\vartheta_{{\overset{ˇ}{c}}_{ij}}\right)}^{{Ω}_{i}}{\left(1-\vartheta_{{\overset{ˇ}{c}}_{\delta }}\right)}^{{Ω}_{i}}\right)}^{\gamma_{j}}\rangle \\ =\langle \mu_{{\overset{ˇ}{c}}_{\delta }}\prod_{j=1}^{m}{\left(\prod_{i=1}^{n}{\left(\mu_{{\overset{ˇ}{c}}_{ij}}\right)}^{{Ω}_{i}}\right)}^{\gamma_{j}},\;1-\left(1-\vartheta_{{\overset{ˇ}{c}}_{\delta }}\right)\prod_{j=1}^{m}{\left(\prod_{i=1}^{n}{\left(1-\vartheta_{{\overset{ˇ}{c}}_{ij}}\right)}^{{Ω}_{i}}\right)}^{\gamma_{j}}\rangle \\ =\langle \prod_{j=1}^{m}{\left(\prod_{i=1}^{n}{\left(\mu_{{\overset{ˇ}{c}}_{ij}}\right)}^{{Ω}_{i}}\right)}^{\gamma_{j}},\;1-\prod_{j=1}^{m}{\left(\prod_{i=1}^{n}{\left(1-\vartheta_{{\overset{ˇ}{c}}_{ij}}\right)}^{{Ω}_{i}}\right)}^{\gamma_{j}}\rangle ⊗\left\langle \mu_{{\overset{ˇ}{c}}_{\delta }},\;\vartheta_{{\overset{ˇ}{c}}_{\delta }}\right\rangle \\ =IFHSWG\;\left(J_{c_{11}},\;J_{{\overset{ˇ}{c}}_{12}},\ldots ,\;J_{{\overset{ˇ}{c}}_{nm}}\right)⊗J_{{\overset{ˇ}{c}}_{\delta }}. \end{array}$$Which completes the proof.  □

#### 3.3.4. (Homogeneity) 

Prove that $IFHSWG\;\left(\alpha J_{{\overset{ˇ}{c}}_{11}},\;\alpha J_{{\overset{ˇ}{c}}_{12}},\ldots ,\;\alpha J_{{\overset{ˇ}{c}}_{nm}}\right)$ = α $IFHSWG\;\left(J_{{\overset{ˇ}{c}}_{11}},\;J_{{\overset{ˇ}{c}}_{12}},\ldots ,\;J_{{\overset{ˇ}{c}}_{nm}}\right)$ for any positive real number α.

**Proof.** Similar to Section 3.2.4. □

## 4. Multi-Criteria Decision-Making Approach under IFHSS Information 

An MCDM technique is constructed here with underdeveloped operators and described as a numerical illustration to demonstrating their competence.

### 4.1. Proposed Approach to Solve the MCDM Problem

Consider Q = $\left\{Q^{1},\;Q^{2},\;Q^{3},\ldots ,\;Q^{s}\right\}$ to be a set of s alternatives and $X=\{X_{1},X_{2},X_{3},\ldots ,X_{n}\}$ to be a set of n experts. The weights of experts are given as Ω = ${\left({Ω}_{1},\;{Ω}_{1},\ldots ,\;{Ω}_{n}\right)}^{T}$ and ${Ω}_{i}>$0, $\sum_{i=1}^{n}{Ω}_{i}$ = 1. Let L = $\left\{c_{1},\;c_{2},\ldots ,\;c_{m}\right\}$ be a set of attributes with their corresponding multi sub-attributes such as $L^{\prime }$ = $\left\{\left(c_{1\rho }\times c_{2\rho }\times \ldots \times c_{m\rho }\right)\;\mathrm{for}\;\mathrm{all}\;\rho \in \left\{1,\;2,\ldots ,\;t\right\}\;\right\}$ with weights γ = ${\left(\gamma_{1\rho },{\;\gamma }_{2\rho },{\;\gamma }_{3\rho },\ldots ,{\;\gamma }_{m\rho }\right)}^{T}$ such as $\gamma_{\rho }$ > 0, $\sum_{\rho =1}^{t}\gamma_{\rho }$ = 1. The components in the collection of sub-attributes are multi-valued; for the sake of accessibility, the components of $L^{\prime }$ can be stated as $L^{\prime }$ = $\left\{{\overset{ˇ}{c}}_{\partial }:\partial \in \left\{1,\;2,\;\ldots ,k\right\}\right\}$. The team of experts {$X_{i}$: i = 1, 2,…, n} appraise the alternatives {$Q^{\left(z\right)}$: z = 1, 2, …, s} under the preferred sub-attributes of the considered parameters {${\overset{ˇ}{c}}_{\partial }$: ∂ = 1, 2, …, k} given in the form of IFHSNs such as ${\left(J_{{\overset{ˇ}{c}}_{ij}}^{\left(z\right)}\right)}_{n\times \partial }$ = ${\left(\mu_{{\overset{ˇ}{c}}_{ij}}^{\left(z\right)},\;\vartheta_{{\overset{ˇ}{c}}_{ij}}^{\left(z\right)}\right)}_{n\times \partial }$, where 0 ≤ $\mu_{{\overset{ˇ}{c}}_{ij}}^{\left(z\right)},\;\vartheta_{{\overset{ˇ}{c}}_{ij}}^{\left(z\right)}$ ≤ 1 and $\mu_{{\overset{ˇ}{c}}_{ij}}^{\left(z\right)}+\vartheta_{{\overset{ˇ}{c}}_{ij}}^{\left(z\right)}$ ≤ 1 for all i, j. 

### 4.2. Algorithm for Developed Aggregation Operators under IFHSS Information

Step 1.Acquire a decision matrix from experts for each alternative {$Q^{\left(z\right)}$: z = 1, 2, …, s} such as follows: $$\begin{array}{l} \begin{array}{cccc} {\overset{ˇ}{c}}_{1} & {\overset{ˇ}{c}}_{2} & \ldots \ldots & \;\;\;\;\;\;\;{\overset{ˇ}{c}}_{\partial } \end{array}\\ {\left(Q^{\left(z\right)},\;L^{\prime }\right)}_{n\times \partial }=\begin{array}{c} X_{1}\\ X_{2}\\ \vdots \\ X_{n} \end{array}(\begin{array}{cccc} \left(\mu_{{\overset{ˇ}{c}}_{11}}^{\left(z\right)},\;\vartheta_{{\overset{ˇ}{c}}_{11}}^{\left(z\right)}\right) & \left(\mu_{{\overset{ˇ}{c}}_{12}}^{\left(z\right)},\;\vartheta_{{\overset{ˇ}{c}}_{12}}^{\left(z\right)}\right) & \cdots & \left(\mu_{{\overset{ˇ}{c}}_{1\partial }}^{\left(z\right)},\;\vartheta_{{\overset{ˇ}{c}}_{1\partial }}^{\left(z\right)}\right)\\ \left(\mu_{{\overset{ˇ}{c}}_{21}}^{\left(z\right)},\;\vartheta_{{\overset{ˇ}{c}}_{21}}^{\left(z\right)}\right) & \left(\mu_{{\overset{ˇ}{c}}_{22}}^{\left(z\right)},\;\vartheta_{{\overset{ˇ}{c}}_{22}}^{\left(z\right)}\right) & \cdots & \left(\mu_{{\overset{ˇ}{c}}_{2\partial }}^{\left(z\right)},\;\vartheta_{{\overset{ˇ}{c}}_{2\partial }}^{\left(z\right)}\right)\\ \vdots & \vdots & \vdots & \vdots \\ \left(\mu_{{\overset{ˇ}{c}}_{n1}}^{\left(z\right)},\;\vartheta_{{\overset{ˇ}{c}}_{n1}}^{\left(z\right)}\right) & \left(\mu_{{\overset{ˇ}{c}}_{n2}}^{\left(z\right)},\;\vartheta_{{\overset{ˇ}{c}}_{n2}}^{\left(z\right)}\right) & \cdots & \left(\mu_{{\overset{ˇ}{c}}_{n\partial }}^{\left(z\right)},\;\vartheta_{{\overset{ˇ}{c}}_{n\partial }}^{\left(z\right)}\right) \end{array}) \end{array}$$Step 2.Transforming cost type sub-attributes to benefit type sub-attribute using the normalization rule and obtain the normalized decision matrix.
$${𝒽}_{ij}=\{\begin{array}{c} J_{{\overset{ˇ}{c}}_{ij}}^{c}=\left(\vartheta_{{\overset{ˇ}{c}}_{ij}}^{\left(z\right)},\;\mu_{{\overset{ˇ}{c}}_{ij}}^{\left(z\right)}\right);\;\;\;\;\;\;\;\mathrm{cost}\;\mathrm{type}\;\mathrm{parameter}\\ J_{{\overset{ˇ}{c}}_{ij}}=\left(\mu_{{\overset{ˇ}{c}}_{ij}}^{\left(z\right)},\;\vartheta_{{\overset{ˇ}{c}}_{ij}}^{\left(z\right)}\right);\;\mathrm{benefit}\;\mathrm{type}\;\mathrm{parameter} \end{array}$$Step 3.Utilizing the developed aggregate, operators obtained a collective decision matrix $J_{{\overset{ˇ}{c}}_{ij}}$ for each alternative T = $\left\{T^{1},\;T^{2},\;T^{3},\ldots ,\;T^{s}\right\}$.Step 4.If T = $\left\{T^{1},\;T^{2},\;T^{3},\ldots ,\;T^{s}\right\}$ are a collection of considered alternatives, then calculate the score values utilizing Equation (1).Step 5.Pick the most suitable alternate with a supreme score value.Step 6.Rank the alternatives.

The above presented approach can be expressed graphically in Figure 1 such as follows:

### 4.3. Case Study

The problem of supplier selection is important in both methodical and practical dimensions. It is a key issue for the firm as the optimal choice of supplier is the basis for effective management of the supply chain and the source of competitive advantage. Including pro-environmental imperatives and other aspects of sustainable development in the supplier selection process makes the correct supplier selection difficult and multidimensional. Depending on the scope of pro-environmental or pro-social activities, supplier selection is often referred to in the literature as “sustainable supplier selection”. It is a complex and multidimensional problem, where there are conflicting criteria, and the evaluation process itself requires consideration of many perspectives. From these points of view, the supplier selection problem is often treated in the literature as a “reference” problem, where multi-criteria decision support methods (MCDM) have been widely applied. The problem of selecting and evaluating a sustainable supplier is addressed in many works. Analyzing papers [49,50,51] and using review papers [52,53,54] in this example of sustainable supplier selection, it was decided to extract the following set of five basic criteria. These are: $c_{1}$, quality of service; $c_{2}$, pollution control; $c_{3}$, environmental efficiency; $c_{4}$_,_ price; and $c_{5}$, corporate social responsibility.

Let {$X^{\left(1\right)}$, $X^{\left(2\right)}$, $X^{\left(3\right)}$, $X^{\left(4\right)}$, $X^{\left(5\right)}$} be a set of substitutes and 𝔏 = $\left\{c_{1}=\mathrm{Superiority},\;c_{2}=\mathrm{Delivery},\;c_{3}=\mathrm{Services},\;c_{4}=\mathrm{Troposphere},\;c_{5}=\mathrm{Commercial}\;\mathrm{societal}\;\mathrm{concern}\right\}$ be a collection of considered attributes given as Superiority = $c_{1}$ = $\left\{c_{11}=national\;level,\;c_{12}=international\;level\right\}$, Delivery = $c_{2}$ = $\left\{c_{21}=by\;carriar,\;c_{22}=by\;hand\right\}$, Services = $c_{3}$ = $\left\{c_{31}=services\right\}$, Troposphere = $c_{4}$ = $\left\{c_{41}=friendly,\;c_{42}=non\;serious\right\}$, and Commercial societal concern = $c_{5}$ = $\left\{c_{51}=\mathrm{Commercial}\;\mathrm{societal}\;\mathrm{concern}\right\}$. Let $L^{\prime }$ = $c_{1}$ × $c_{2}$ × $c_{3}$× $c_{4}$ × $c_{5}$ be a set of sub-attributes
$$\begin{array}{l} L^{\prime }=c_{1}\;\times \;c_{2}\;\times \;c_{3}\times \;c_{4}\;\times \;c_{5}=\left\{c_{11},\;c_{12}\right\}\times \left\{c_{21},\;c_{22}\right\}\times \left\{c_{31},\;c_{32}\right\}\times \left\{c_{41}\right\}\times \left\{c_{51}\right\}\\ =\left\{\begin{array}{c} \left(c_{11},\;c_{21},\;c_{31},\;c_{41},\;c_{51}\right),\;\left(c_{11},\;c_{21},\;c_{32},\;c_{41},\;c_{51}\right),\;\left(c_{11},\;c_{22},\;c_{31},\;c_{41},\;c_{51}\right),\;\left(c_{11},\;c_{22},\;c_{32},\;c_{41},\;c_{51}\right),\\ \left(c_{12},\;c_{21},\;c_{31},\;c_{41},\;c_{51}\right),\;\left(c_{12},\;c_{21},\;c_{32},\;c_{41},\;c_{51}\right),\;\left(c_{12},\;c_{22},\;c_{31},\;c_{41},\;c_{51}\right),\;\left(c_{12},\;c_{22},\;c_{32},\;c_{41},\;c_{51}\right) \end{array}\right\},\; \end{array}$$

$L^{\prime }$ = $\left\{{\overset{ˇ}{c}}_{1},{\overset{ˇ}{c}}_{2},\;{\overset{ˇ}{c}}_{3},\;{\overset{ˇ}{c}}_{4},\;{\overset{ˇ}{c}}_{5},\;{\overset{ˇ}{c}}_{6},\;{\overset{ˇ}{c}}_{7},\;{\overset{ˇ}{c}}_{8}\right\}$ be a set of all sub-attributes with weights ${\left(0.12,\;0.18,\;0.1,0.15,\;0.05,\;0.22,\;0.08,\;0.1\right)}^{T}$. Consider {$Q^{\left(1\right)}$, $Q^{\left(2\right)}$, $Q^{\left(3\right)}$, $Q^{\left(4\right)}$} be a set of experts with weights ${\left(0.2,\;0.3,\;0.4,\;0.1\right)}^{T}$ to assess the finest alternate. Experts give their preferences in terms of IFHSNs using considered multi sub-attributes. The following is that the procedure progressed to get the most productive choice.

### 4.4. Sustainable Supplier Selection Using Intuitionistic Fuzzy Hypersoft Weighted Average Operator

Step 1.Experts examine the circumstances in the instance of IFHSNs. The multi sub-attributes of the considered attributes, along with a summary of their score values are given in Table 1, Table 2, Table 3 and Table 4.Step 2.There is no need for normalization, as all attributes are identical.Step 3.Using Equation (4), expert’s opinion can be summarized as follows:${ℒ}_{1}$ = ⟨0.23054, 0.42563⟩, ${ℒ}_{2}$ = ⟨0.291197, 0.415560⟩, ${ℒ}_{3}$ = ⟨0.447878, 0.363226⟩, and ${ℒ}_{4}$ = ⟨0.441142, 0.345641⟩.Step 4.Calculate the score values using Equation (1).$S\left({ℒ}_{1}\right)$ = −0.195086, $S\left({ℒ}_{2}\right)$ = −0.124363, $S\left({ℒ}_{3}\right)$ = 0.084652, and $S\left({ℒ}_{4}\right)$ = 0.095501.Step 5.By using the obtained score values, we can see that $Q^{\left(4\right)}$ is the best alternative.Step 6.Hence, the following is an arrangement of alternatives: $S\left({ℒ}_{4}\right)$ > $S\left({ℒ}_{3}\right)$ > $S\left({ℒ}_{2}\right)$ > $S\left({ℒ}_{1}\right)$. So, $Q^{\left(4\right)}$ > $Q^{\left(3\right)}>Q^{\left(2\right)}$ > $Q^{\left(1\right)}$, hence, the alternative $Q^{\left(4\right)}$ is the most suitable alternative.

### 4.5. Sustainable Supplier Selection Using the Intuitionistic Fuzzy Hypersoft Weighted Geometric Operator

Step 1.Experts examine the circumstances in the instance of IFHSNs. The multi sub-attributes of the considered attributes, along with a summary of their score values are given in Table 1, Table 2, Table 3 and Table 4.Step 2.There is no need for normalization, as all attributes are identical.Step 3.Using Equation (8), expert’s opinion can be summarized as follows:${ℒ}_{1}$ = ⟨0.201387, 0.461254⟩, ${ℒ}_{2}$ = ⟨0.238098, 0.480574⟩, ${ℒ}_{3}$ = ⟨0.317227, 0.459177⟩, and ${ℒ}_{4}$ = ⟨0.366312, 0.402225⟩.Step 4.Calculate the score values using Equation (1).$S\left({ℒ}_{1}\right)$ = −0.259867, $S\left({ℒ}_{2}\right)$ = −0.242376, $S\left({ℒ}_{3}\right)$ = −0.141950, and $S\left({ℒ}_{4}\right)$ = −0.035913.Step 5.By using the obtained score values, we can see that $Q^{\left(4\right)}$ is the best alternative.Step 6.Hence, the following is an arrangement of alternatives $S\left({ℒ}_{4}\right)$ > $S\left({ℒ}_{3}\right)$ > $S\left({ℒ}_{2}\right)$ > $S\left({ℒ}_{1}\right)$. So, $Q^{\left(4\right)}$ > $Q^{\left(3\right)}>Q^{\left(2\right)}$ > $Q^{\left(1\right)}$, therefore, the alternate $Q^{\left(4\right)}$ is the most appropriate alternative.

## 5. Comparative Analysis and Discussion

In the subsequent section, we are going to talk over the usefulness, easiness, and manageability of the assistance of the planned method. We also performed an ephemeral evaluation of the undermentioned: the planned technique, along with some prevailing methodologies.

### 5.1. Superiority of the Proposed Method

Through this study and comparison, it could be determined that the consequences acquired by the suggested approach have been more common than either available methods. However, overall, the DM procedure associated with the prevailing DM methods accommodates extra information to address hesitation. In addition, FS’s various hybrid structures are becoming a special feature of IFHSS, along with some appropriate circumstances which have been added. The general information associated with the object can be stated precisely and analytically (see Table 5). Therefore, it is a suitable technique to syndicate inaccurate and ambiguous information in the DM process. Hence, the suggested approach is practical, modest, and in advance of fuzzy sets’ distinctive hybrid structures.

### 5.2. Discussion

Using MD, Zadeh’s [1] FS only handled the inexact and imprecise information of sub-attributes of considered attributes for each alternative. However, the FS has no evidence regarding the NMD of the considered parameters. The existing FS only contracts the ambiguous difficulties using MD, though our planned technique accommodates the vagueness applying MD and NMD. Additionally, Zhang et al. [55] and Xu et al. [56] developed IFS which deals with the vague information using MD and NMD. However, these theories are unable to deal with the parametric values of the alternatives. Maji et al. [7] presented the notion of FSS to deal with the parametrization of the objects which contain uncertainty by considering the MD of the attributes. However, the presented FSS provides no information about the NMD of the object. To overcome the presented drawback, Maji et al. [9] offered the concept of IFSS; the presented notion handles the uncertain object more accurately, utilizing the MD and NMD of the attributes with their parametrization and MD+ NMD≤1. All the above mentioned studies have no information about the sub-attributes of the considered attributes. Therefore, the above mentioned theories are unable to handle the scenario when attributes have their corresponding sub-attributes. Our presented approach can resolve these complications simply, and bring additional operational consequences in the DM method. It may be seen that the best selection of the suggested approach is to resemble the verbalized own method, and this ensures the liableness along with the effectiveness of the recommended approach.

### 5.3. Comparative Analysis

We recommend another algorithmic rule under IFHSS by utilizing the established IFHSWA and IFHSWG operators in the following section. Subsequently, we utilize the suggested algorithm to a realistic problem, namely the supplier selection in SSCM. The overall outcomes prove that the algorithmic rule is valuable and practical. It can be observed that Q^(4^^)^ supplier is the finest alternative for SSCM (see Table 6). The recommended approach may be compared to other available methods. From the research findings, it has been concluded that the outcomes acquired by the planned approach exceed the consequences of the prevailing ideas. Therefore, comparative to existing techniques, the established AOs competently handled the uncertain and ambiguous information. However, under existing DM strategies, the core advantage of the planned method is that it can accommodate extra information in data comparative to existing techniques. It is also a beneficial tool to solve inaccurate, as well as imprecise, information in DM procedures. 

The existing operators are unable to cope with parametrization and multi sub-attributes of the alternatives. However, our developed IFHSWA and IFHSWG operators competently handle the parametrization and multi sub-attributes of the considered attributes. Therefore, it is a suitable tool to merge inexact as well as hesitant information in the DM method.

## 6. Conclusions

This study focuses on IFHSS to resolve the problems of inadequate, indistinct, and discrepant information by considering the MD and NMD on the n-tuple sub-attributes of the considered attributes. This research puts forward the novel aggregation operators for IFHSS, such as IFHSWA and IFHSWG operators with their fundamental properties. Additionally, a DM approach has been developed by using IFHSWA or IFHSWG operators to solve MCDM complications. Furthermore, a comparative analysis was performed to ensure the effectiveness and manifestation of the presented approach. Lastly, according to the obtained outcomes, it can be determined that the planned method displays advanced constancy and feasibility for experts in the DM procedure. Based on the obtained results, it can be determined that the planned technique shows that the experts have higher stability and availability in the DM process. The forthcoming study will focus on presenting DM techniques utilizing numerous other measures, such as entropy measures and similarity measures, etc., under IFHSS. Moreover, many other structures can be established and proposed, such as topological structure, algebraic structure, ordered structure, etc. We believe that this research opens new vistas for investigators in this field.

## Figures and Tables

**Figure 1 entropy-23-00688-f001:**
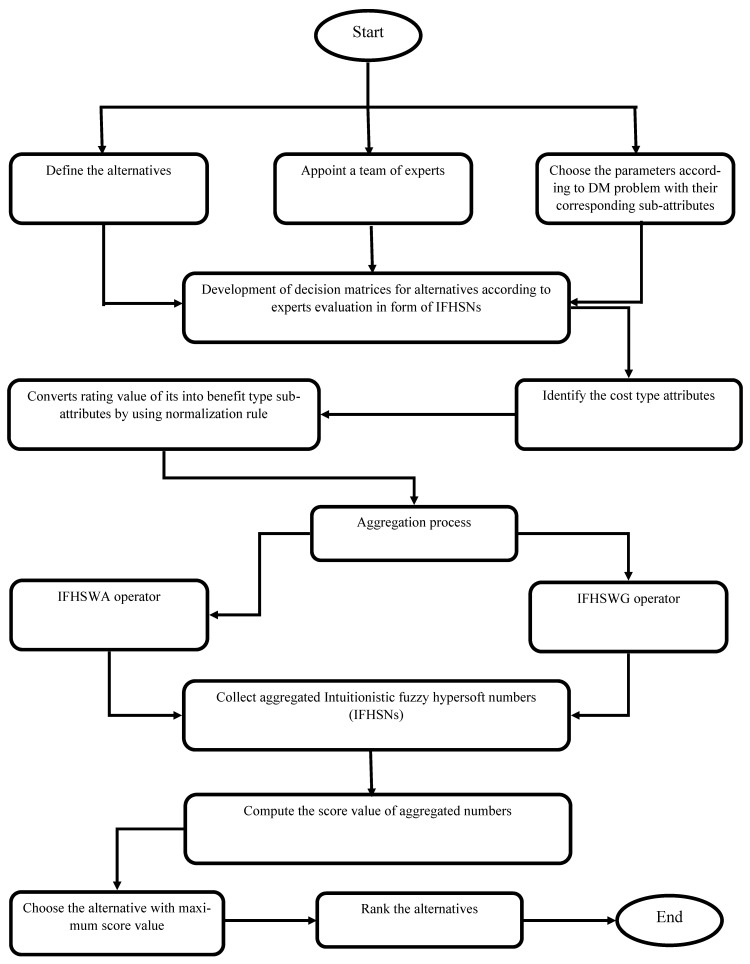
Flow chart of the presented decision-making approach.

**Table 1 entropy-23-00688-t001:** Decision Matrix for Alternative $Q^{\left(1\right)}$.

$Q^{\left(1\right)}$	${\overset{ˇ}{c}}_{1}$	${\overset{ˇ}{c}}_{2}$	${\overset{ˇ}{c}}_{3}$	${\overset{ˇ}{c}}_{4}$	${\overset{ˇ}{c}}_{5}$	${\overset{ˇ}{c}}_{6}$	${\overset{ˇ}{c}}_{7}$	${\overset{ˇ}{c}}_{8}$
$X_{1}$	(0.3, 0.5)	(0.2, 0.3)	(0.1, 0.3)	(0.3, 0.6)	(0.2, 0.4)	(0.2, 0.6)	(0.5, 0.4)	(0.1, 0.3)
$X_{2}$	(0.2, 0.7)	(0.4, 0.6)	(0.3, 0.4)	(0.1, 0.2)	(0.1, 0.2)	(0.2, 0.4)	(0.2, 0.5)	(0.4, 0.5)
$X_{3}$	(0.2, 0.3)	(0.2, 0.5)	(0.1, 0.6)	(0.3, 0.4)	(0.4, 0.6)	(0.1, 0.4)	(0.2, 0.3)	(0.2, 0.5)
$X_{4}$	(0.2, 0.4)	(0.2, 0.3)	(0.2, 0.4)	(0.4, 0.6)	(0.3, 0.5)	(0.3, 0.6)	(0.4, 0.5)	(0.1, 0.3)

**Table 2 entropy-23-00688-t002:** Decision Matrix for Alternative $Q^{\left(2\right)}$.

$Q^{\left(2\right)}$	${\overset{ˇ}{c}}_{1}$	${\overset{ˇ}{c}}_{2}$	${\overset{ˇ}{c}}_{3}$	${\overset{ˇ}{c}}_{4}$	${\overset{ˇ}{c}}_{5}$	${\overset{ˇ}{c}}_{6}$	${\overset{ˇ}{c}}_{7}$	${\overset{ˇ}{c}}_{8}$
$X_{1}$	(0.2, 0.6)	(0.3, 0.4)	(0.4, 0.5)	(0.3, 0.5)	(0.5, 0.4)	(0.4, 0.6)	(0.3, 0.5)	(0.4, 0.5)
$X_{2}$	(0.3, 0.5)	(0.2, 0.4)	(0.1, 0.2)	(0.1, 0.2)	(0.4, 0.5)	(0.1, 0.3)	(0.2, 0.7)	(0.1, 0.8)
$X_{3}$	(0.3, 0.7)	(0.4, 0.5)	(0.2, 0.8)	(0.3, 0.4)	(0.2, 0.3)	(0.3, 0.4)	(0.1, 0.2)	(0.7, 0.2)
$X_{4}$	(0.5, 0.4)	(0.1, 0.6)	(0.2, 0.3)	(0.2, 0.3)	(0.1, 0.2)	(0.2, 0.4)	(0.4, 0.6)	(0.5, 0.5)

**Table 3 entropy-23-00688-t003:** Decision Matrix for Alternative $Q^{\left(3\right)}$.

$Q^{\left(3\right)}$	${\overset{ˇ}{c}}_{1}$	${\overset{ˇ}{c}}_{2}$	${\overset{ˇ}{c}}_{3}$	${\overset{ˇ}{c}}_{4}$	${\overset{ˇ}{c}}_{5}$	${\overset{ˇ}{c}}_{6}$	${\overset{ˇ}{c}}_{7}$	${\overset{ˇ}{c}}_{8}$
$X_{1}$	(0.4, 0.5)	(0.3, 0.5)	(0.4, 0.5)	(0.3, 0.4)	(0.2, 0.4)	(0.4, 0.5)	(0.3, 0.4)	(0.3, 0.5)
$X_{2}$	(0.3, 0.4)	(0.1, 0.3)	(0.1, 0.8)	(0.1, 0.2)	(0.4, 0.6)	(0.3, 0.7)	(0.6, 0.1)	(0.8, 0.1)
$X_{3}$	(0.6, 0.2)	(0.3, 0.4)	(0.7, 0.3)	(0.3, 0.4)	(0.1, 0.2)	(0.4, 0.5)	(0.3, 0.5)	(0.6, 0.3)
$X_{4}$	(0.5, 0.4)	(0.2, 0.3)	(0.4, 0.6)	(0.3, 0.4)	(0.3, 0.6)	(0.7, 0.2)	(0.4, 0.2)	(0.5, 0.2)

**Table 4 entropy-23-00688-t004:** Decision Matrix for Alternative $Q^{\left(4\right)}$.

$Q^{\left(4\right)}$	${\overset{ˇ}{c}}_{1}$	${\overset{ˇ}{c}}_{2}$	${\overset{ˇ}{c}}_{3}$	${\overset{ˇ}{c}}_{4}$	${\overset{ˇ}{c}}_{5}$	${\overset{ˇ}{c}}_{6}$	${\overset{ˇ}{c}}_{7}$	${\overset{ˇ}{c}}_{8}$
$X_{1}$	(0.2, 0.7)	(0.4, 0.5)	(0.2, 0.4)	(0.4, 0.3)	(0.1, 0.2)	(0.2, 0.4)	(0.3, 0.4)	(0.2, 0.4)
$X_{2}$	(0.3, 0.5)	(0.2, 0.4)	(0.8, 0.1)	(0.5, 0.2)	(0.4, 0.3)	(0.4, 0.5)	(0.7, 0.2)	(0.6, 0.3)
$X_{3}$	(0.6, 0.3)	(0.4, 0.5)	(0.6, 0.2)	(0.6, 0.4)	(0.1, 0.2)	(0.3, 0.4)	(0.5, 0.3)	(0.4 0.5)
$X_{4}$	(0.5, 0.4)	(0.1, 0.3)	(0.3, 0.5)	(0.5, 0.3)	(0.3, 0.5)	(0.8, 0.1)	(0.3, 0.5)	(0.2, 0.5)

**Table 5 entropy-23-00688-t005:** Comparison of IFHSSs with some prevailing studies.

	Set	Truthiness	Falsity	Attributes	Sub-Attributes	Loss of Information	Parametrization	Advantages
Zadeh [1]	FS	✓	×	✓	×	×	×	Deals uncertainty by using fuzzy interval
Maji et al. [7]	FSS	✓	×	✓	×	×	✓	Deals uncertainty by using fuzzy soft intervals
Zhang et al. [55]	IFS	✓	✓	✓	×	✓	×	Deals uncertainty by using MD and NMD
Xu et al. [56]	IFS	✓	✓	✓	×	×	×	Deals uncertainty by using MD and NMD
Maji et al. [9]	IFSS	✓	✓	✓	×	×	✓	Deals uncertainty by using MD and NMD
Proposed approach	IFHSS	✓	✓	✓	✓	×	✓	Deals more uncertainty comparative to IFHSS

**Table 6 entropy-23-00688-t006:** Comparative analysis with existing operators.

Method	$Q^{\left(1\right)}$	$Q^{\left(2\right)}$	$Q^{\left(3\right)}$	$Q^{\left(4\right)}$	Ranking Order
IFWA [25]	0.21173	0.22017	0.33215	0.27008	$Q^{\left(4\right)}$ > $Q^{\left(1\right)}>Q^{\left(2\right)}$ > $Q^{\left(3\right)}$
IFWG [21]	0.20587	0.23066	0.32902	0.25462	$Q^{\left(4\right)}$ > $Q^{\left(1\right)}>Q^{\left(2\right)}$ > $Q^{\left(3\right)}$
IFEWA [22]	0.51686	0.54833	0.60467	0.59021	$Q^{\left(4\right)}$ > $Q^{\left(1\right)}>Q^{\left(2\right)}$ > $Q^{\left(3\right)}$
IFEWG [20]	0.54219	0.56597	0.62190	0.59381	$Q^{\left(4\right)}$ > $Q^{\left(1\right)}>Q^{\left(2\right)}$ > $Q^{\left(3\right)}$
IFSWA [13]	0.08158	0.07674	0.14762	0.09959	$Q^{\left(4\right)}$ > $Q^{\left(1\right)}>Q^{\left(2\right)}$ > $Q^{\left(3\right)}$
IFSWG [13]	0.49830	0.41735	0.40935	0.46175	$Q^{\left(4\right)}$ > $Q^{\left(1\right)}>Q^{\left(2\right)}$ > $Q^{\left(3\right)}$
Proposed IFHSWA operator	−0.195086	−0.124363	0.084652	0.095501	$Q^{\left(4\right)}$ > $Q^{\left(3\right)}>Q^{\left(2\right)}$ > $Q^{\left(1\right)}$
Proposed IFHSWG operator	−0.259867	−0.242376	−0.141950	−0.035913	$Q^{\left(4\right)}$ > $Q^{\left(3\right)}>Q^{\left(2\right)}$ > $Q^{\left(1\right)}$

## Data Availability

Not applicable.

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
