# Peer review of "Robust Aggregation Operators for Intuitionistic Fuzzy Hypersoft Set with Their Application to Solve MCDM Problem"

_entropy, 2021, doi:10.3390/e23060688_

Round 1

Reviewer 1 Report

The focal topic in the paper is interesting and of significance. However, currently, the manuscript must be go through major revision according to following suggestions:

  1. The introduction part was too concise to show the motivations and significance of the proposed decision making tools.
  2. The Robustness of the proposed aggregation operators must be clearly addressed.
  3. Practical examples must be presented to illustrate the superiority of the proposed fuzzy tools.
  4. More comparative studies should be conducted to demonstrate the effectiveness and advantages of the proposed decision making approach.

5. More aggregation operators should be further discussed to enhance the applicability of the research job. 

Author Response

Dear Respected Reviewer,

We are thankful to you for your positive comments to improve our manuscript. We revised the manuscript according to the valuable suggestions and highlights the changes with color (Blue). Hopefully, our modifications in the revised manuscript would achieve your acceptance. Please see the revised version of the manuscript.

Reviewer 2 Report

The authors propose aggregation operators for the intuitionistic fuzzy hypersoft set, which is the generalization and extension of the intuitionistic fuzzy soft set. The paper presents operational laws for intuitionistic fuzzy hypersoft numbers and aggregation operators. Tha authors also proposed a decision-making approach. The paper is a interesting scientific work. However some shortcomings must be eliminated before accepting. The list of comments is as follows:
1. The abstract must be rewritten. There should be clear emphasis motivation and contribution.
2. Introduction must be extended. Authors omitted current MCDA  trends. Must be presented additional background, e.g., Normalized weighted Geometric Dombi Bonferoni Mean Operator with interval grey numbers: Application in multicriteria decision making; Best-Worst method and Hamacher aggregation operations for intuitionistic 2-tuple linguistic sets; A new method to support decision-making in an uncertain environment based on normalized interval-valued triangular fuzzy numbers and comet technique; or similar 
3. What is the relationship this paper with entropy? It is not presented in abstract and in Introduction.
4. Sections 3 5 should be extended. Now, it is too short. Authors should extend it, e.g., line 172 it is only one description in section 3.
5. The paper has a lot of borrowing from work Aggregation operators of Pythagorean fuzzy soft sets with their application for green supplier chain management; and from http://fs.unm.edu/NSS/ADevelopmentOfPythagoreanFuzzyHypersoft9.pdf and from TOPSIS Method Based on Correlation Coefficient under Pythagorean Fuzzy Soft Environment and Its Application towards Green Supply Chain Management
6. Comparative Analysis is not enough form section 5.2. It must be extended
7. The algorithms in section 4.4 and 4.5 should be presented as flowchart (additionally).
8. The further works must be presented in conclusions.
9. The proposed algorithms must be presented with using non trivial examples and similar approaches to compare the efficiency of the proposed approaches and operators. 

Author Response

(The authors gave the same response as above.)

Round 2

Reviewer 2 Report

The paper has been improved. I suggest accepting in the current form.